# Beyond Literal Translation: Evaluating Cultural Effectiveness in Social Media UGC

**Linjuan Wu** [* 1 2] **Ruiqi Zhang** [* 3 2] **Xinze Lyu** [2] **Ye Guo** [2] **Daoxin Zhang** [2] **Zhe Xu** [2] **Yao Hu** [2] **Yixin Cao** [3] **Yongliang Shen** [1] **Weiming Lu** [1]

## Abstract

Social media platforms enable large-scale cross-lingual communication, but translating user-generated content (UGC) remains challenging due to its informal style, cultural references, and interaction-based expressions. While recent LLMs have improved translation quality, existing benchmarks and metrics often fail to capture whether translations convey intended meaning and cultural resonance in real-world settings. In this work, we introduce **CULTURE-MT**, a benchmark for social media translation that focuses on both **CUL**tural **T**ransmission and **U**GC-specific emotion **RE**sonance. CULTURE-MT consists of 1,002 UGC notes across 14 domains, categorized into four types based on culture-loaded symbol and linguistic style features. We also construct UGC-oriented training data to fine-tune Qwen3-8B and Qwen3-32B as baselines. We propose **cultural effectiveness** as a new evaluation criterion, focusing on expression accuracy and cultural adaptability. Testing 15 models, including the baselines, we find that traditional metrics fail to capture cultural effectiveness. We also observe that cultural effectiveness on base LLMs correlates with model size. Our work provides a comprehensive evaluation system for UGC translation models and will offers an open evaluation platform to advance research in this area. We release the CULTURE-MT benchmark and provide an online leaderboard where submitted translation results can be evaluated by our trained JUDGER.

---

[*]Equal contribution . This work was completed while Linjuan Wu and Ruiqi Zhang were interns at Xiaohongshu Inc. Contact: ⟨wulinjuan525@zju.edu.cn⟩ ⟨24210240070@m.fudan.edu.cn⟩ [1]Zhejiang University [2]Xiaohongshu Inc. [3]Fudan University. Correspondence to: Weiming Lu <luwm@zju.edu.cn>, Daoxin Zhang <tangxiaohui@xiaohongshu.com>.

*Proceedings of the 43rd International Conference on Machine Learning*, Seoul, South Korea. PMLR 306, 2026. Copyright 2026 by the author(s).

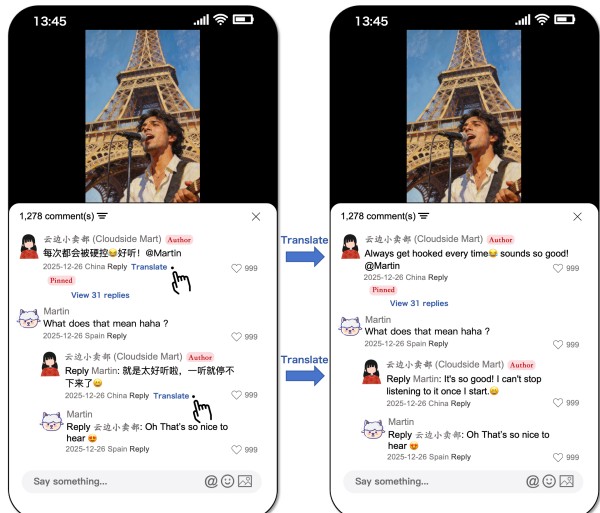

*Figure 1.* Example of cross-lingual interaction on a social platform via automatic translation. A Chinese user's comment with buzzwords is translated to English, but the literal translation fails to convey the intended meaning. This illustrates the challenges of translating context-rich, culturally specific UGC on social media.

## 1. Introduction

Social media platforms have transformed how people communicate, and access information globally. User-generated content (UGC), such as posts, comments, and personal notes, is a key way for individuals to explore different lifestyles, values, and cultures (Lin et al., 2018; Chouaki et al., 2024; Vombatkere et al., 2024; Jin et al., 2024; Wei et al., 2025; Kim & Introne, 2025; Ye & Gao, 2026; Liu et al., 2025; Rehman et al., 2026). However, online communities remain divided by language and culture, with most users engaging mainly with content in their native language. This limits cross-lingual information exchange and cross-cultural understanding(Guo et al., 2025a).

With advancements in LLMs (Google, 2025; Team et al., 2025; OpenAI, 2025; Yang et al., 2025; DeepSeek-AI et al., 2025; Grattafiori et al., 2024), several platforms have integrated machine translation (MT) based on it to enable cross-lingual understanding and communication (Figure 1). However, deploying large-scale or closed-source LLMs is

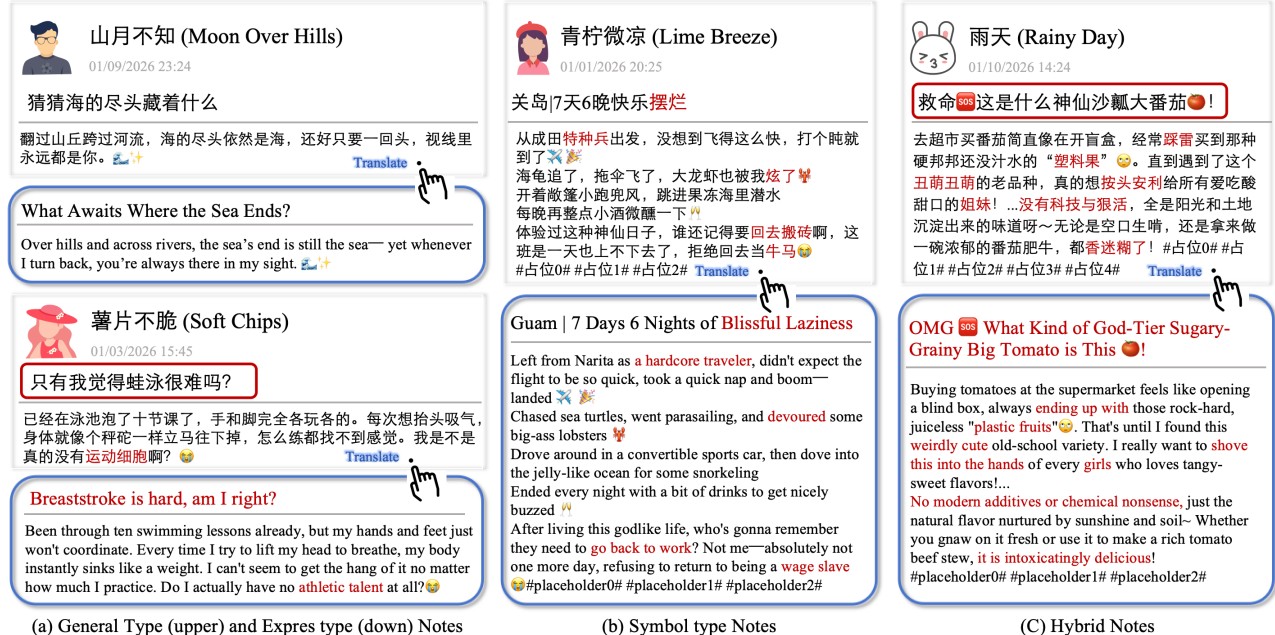

*Figure 2.* Representative examples of different combinations of language-loaded symbols and linguistic styles in Chinese UGC: (a) General Note with informal expression and few cultural-loaded symbols, and Express Note with unique linguistic styles; (b) Stylistic Note with rich culture-loaded symbols; (c) Hybrid Note with both characteristics. In addition to culture-loaded symbols, UGC often features rhetorical, expressive, and interactive language styles, which existing benchmarks do not fully address, posing new challenges for social media translation.

not feasible for most platforms due to their high computational costs. The translations system utilized in platforms still tend to literal approaches, which fail to capture the nuanced meaning of informal, context-heavy, and culturally specific UGC (Macko et al., 2025; Huang et al., 2025; Mei et al., 2024; Wuraola et al., 2024; Zhang et al., 2023). There is a growing need for more efficient translation models that strike a balance between high-quality output and scalability, allowing for continuous iteration and widespread deployment on social media platforms.

Several benchmarks have been proposed to develop and evaluate social media translation models. Redtrans-Bench (Guo et al., 2025b), Seed-X-Challenge (Cheng et al., 2025), and DITING-Corpus (Zhang et al., 2025) address various cultural challenges in online language, including humor localization, slang, and idiomatic expressions. These benchmarks highlight the culture-loaded symbols (e.g., as shown in Figure 2b), they overlook the unique linguistic styles in UGC shaped by social interaction. For example, rhetorical questions are used to prompt agreement (Figure 2a), or exaggerated language to attract attention (Figure 2c). Translating culture-loaded symbols ensures semantic accuracy, while translating linguistic style is crucial for evoking emotional resonance and sustaining interaction. Thus, a benchmark framework that evaluates both culture-loaded symbols and UGC-specific linguistic styles is essential for advancing translation systems in real-world social media contexts.

Motivated by the above, we construct **CULTURE-MT**, a benchmark designed to evaluate translation models for Chinese-to-English UGC Notes. CULTURE-MT consists of 1,002 UGC Notes across 14 content domains, explicitly accounting for both culture-loaded symbols and UGC-specific linguistic styles. Based on the presence of these two aspects, we categorize the Notes into four types—General, Symbol, Express, and Hybrid, as shown in Figure 2—encompassing a wide range of linguistic and stylistic phenomena commonly found on Chinese social media platforms. We scope CULTURE-MT to self-contained, text-evaluable notes: content that requires substantial image/video evidence, thread history, or highly localized external context is treated as an important but separate setting for future contextual and multimodal evaluation.

To complete the evaluation framework, we propose a new evaluation criterion, **cultural effectiveness**, to assess how well translations convey the intended meaning and evoke the appropriate cultural resonance. This criterion complements traditional metrics by focusing on both expression accuracy and cultural adaptability. It can be viewed as a task-specific MQM-style (Lommel et al., 2014) rubric tailored to UGC, where cultural term handling, pragmatic intent, discourse style, and target-reader interpretability are made explicit. To enable large-scale evaluation, we train an automatic evaluator, JUDGER, using expert-annotated and LLM-generated UGC translation samples. We validate its reliability by com-

paring accuracy (Acc) and Cohen's Kappa coefficient with human expert scores, showing strong agreement. Furthermore, we synthesize UGC translation data using an LLM and fine-tune two translation models based on Qwen3-32B and Qwen3-8B as strong baselines for this benchmark.

We evaluate 15 models—including UGC translation baselines—on CULTURE-MT, reporting cultural effectiveness alongside BLEU, ChrF, and COMET. Among 1,002 instances, the best-performing model (Gemini 3) achieves only 38.30% top-rated culturally effective translations, versus 28.24% for an 8B baseline, highlighting the benchmark's challenge. While cultural effectiveness shows a clear scaling trend with model size, standard metrics like BLEU and ChrF fail to reflect these meaningful differences, underscoring their insensitivity to cultural nuance.

We make three key contributions:

- We introduce **CULTURE-MT**, a challenging benchmark for Chinese-to-English UGC translation that emphasizes culture-loaded expressions and social media–specific linguistic styles.

- We propose **cultural effectiveness** as a new evaluation criterion that captures both translation accuracy and cultural resonance, addressing a critical limitation of standard automatic metrics.

- Through large-scale evaluation of 15 models, we reveal a strong correlation between model scale and cultural effectiveness, and release Qwen3-8B and Qwen3-32B as **strong baselines** to advance research in this direction.

## 2. Related Work

### 2.1. Translation Benchmark for Online Content

As general-purpose translation capabilities advance, researchers are increasingly focusing on translating online content to promote cross-cultural communication (Zhao et al., 2025b;a; Li et al., 2025; Guo et al., 2025a; Feng et al., 2025a;b). The Seed-X-Challenge (Cheng et al., 2025) is a multilingual benchmark that covers colloquial and slang-heavy internet text across multiple domains from online platforms. The most closely related benchmark to our work is RedTrans-Bench (Guo et al., 2025b), which evaluates LLMs on cross-cultural transfer in social media contexts, specifically targeting humor localization, emoji semantics, and meme adaptation. While these are important cultural challenges on social platforms, their scope is limited, particularly in capturing the diverse linguistic styles found in user-generated content (UGC), such as hyperbole, rhetorical questions, and the varied expressions central to Chinese social media.

Benchmarks like TransBench (Li et al., 2025) focus on professional domains such as e-commerce, legal, and finance, emphasizing accurate terminology and cultural nuance. However, these texts are formal and structured, contrasting sharply with the informal, fragmented nature of social media UGC. Additionally, TransBench has not yet released its data, limiting direct comparison. DITING-Corpus (Zhang et al., 2025) offers an evaluation framework for web-novel translation, defining tasks like idiom translation, lexical ambiguity, and cultural safety. While insightful, DITING-Corpus focuses on long-form narrative text, whereas social media posts present multiple challenges in a single short passage, requiring a more integrated evaluation approach. Specialized efforts like SlangDIT (Liang et al., 2025) and SLANG (Mei et al., 2024) target slang translation, and recent studies (e.g., Wuraola et al. (2024); Huang et al. (2025)) explore LLMs' understanding of internet buzzwords or slang. Benchmarking Machine Translation with Cultural Awareness (Yao et al., 2024), further demonstrate the importance of culture-sensitive evaluation, focusing on their pragmatic translation quality. These works highlight lexical understanding limitations but rarely assess whether translations evoke the intended emotional or social resonance—an essential aspect of cultural effectiveness in our approach.

Recent efforts to evaluate translation with cultural awareness have primarily focused on specific phenomena or narrow verticals, leaving the diverse and complex nature of social media UGC underexplored.

### 2.2. LLM-as-a-Judge for Translation Evaluation

Traditional automatic metrics, such as BLEU, chrF, and COMET, are ill-suited for evaluating translations that must meet domain-specific or culturally nuanced requirements. Cheng et al. (2025) employs human expert evaluation to assess translation quality, but this approach is costly and inefficient at scale. These limitations have spurred growing interest in alternative evaluation paradigms, particularly LLM-based evaluators, which can better capture the subtleties of high-quality translation in specialized contexts.

Recent work increasingly uses LLMs as effective evaluators. For example, DITING (Zhang et al., 2025) introduces AgentEval, a multi-agent framework where two evaluators independently assign scores, and a third arbiter resolves disagreements through iterative debate. Using MetricAlign to evaluates the consistency between automatic metrics and expert judgments. It works by sampling data from each task (12 sentences per task, totaling 300 translations across 25 models), and having experts evaluate these translations. The evaluation includes measuring inter-annotator agreement using Accuracy and Cohen's Kappa coefficient (Cohen's $\kappa$). In domain adaptation, TransBench (Li et al., 2025) introduces Marco-MOS, a quality estimator fine-tuned on 35,000 human-rated translations for e-commerce content.

Trained on the Qwen-Instruct model, Marco-MOS achieves a Pearson correlation of 0.65 with human judgments, outperforming GPT-4 and COMET.

Building on these advances, we introduce a novel evaluation criterion for social media translation: cultural effectiveness. A closely related line of work operationalizes translation quality through MQM-style taxonomies and LLM-based judgments, including MQM (Lommel et al., 2014), GEMBA-MQM (Kocmi & Federmann, 2023), Auto-MQM (Fernandes et al., 2023), MQM-APE (Lu et al., 2025), and broader studies of LLM-as-a-judge reporting and bias (Zheng et al., 2023). We treat MQM as a strong general framework that should be specialized when the target construct is cultural effectiveness. CULTURE-MT therefore makes UGC-specific cultural symbols, pragmatic intent, expressive style, and target-reader resonance explicit in both the human rubric and the trained JUDGER. To operationalize this, we create dedicated training and test sets annotated with cultural effectiveness scores, which we use to develop and validate our LLM-based evaluator, JUDGER, designed to assess whether a translation resonates with target-language users reliably.

**In summary**, our work presents the first multi-domain, multi-format benchmark for social media UGC translation, which not only incorporates buzzwords and slang but also evaluates whether translations engage and resonate with target-language users. Combined with our automatic evaluator (JUDGER), this benchmark offers practical guidance for improving the social-media translation performance of small-scale LLMs, bridging a critical gap in the evaluation–training loop for culturally aware machine translation.

## 3. CULTURE-MT Benchmark

We introduce CULTURE-MT, a Chinese-English translation benchmark for social platform-style user-generated content (UGC). It consists of 1,002 cases spanning 14 verticals, each containing four types of Notes.

### 3.1. Data Categories

**Vertical Field Selection**   With the advancement of globalization, social media platforms are increasingly facilitating communication between users from different countries within the same online community. Content creators on these platforms also aim to attract the attention of international audiences. Based on data from Chinese social platforms, we conducted a survey of vertical domains that attract attention from foreign users and analyzed highly praised posts. As a result, we identified the following 14 verticals, which represent topics that are particularly favored by non-Chinese users: *Pets, Travel, Food, Crafts, Painting, Home Decoration, Outdoor, Sports, Fitness & Weight Loss,*

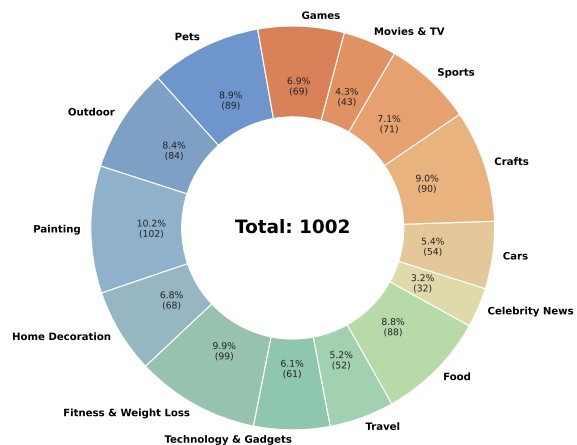

*Figure 3.* Distribution of user verticals across 1,002 annotated samples. Each segment represents a distinct vertical, labeled with its name, percentage share, and absolute count.

*Table 1.* Dataset Statistics of CULTURE-MT Benchmark.

| Category | General | Express | Symbol | Hybrid | Total |
|---|---|---|---|---|---|
| Pets | 10 | 20 | 24 | 35 | 89 |
| Outdoor | 10 | 20 | 28 | 26 | 84 |
| Painting | 13 | 29 | 41 | 19 | 102 |
| Home Decoration | 9 | 13 | 32 | 14 | 68 |
| Fitness & Weight Loss | 13 | 21 | 39 | 26 | 99 |
| Technology & Gadgets | 2 | 20 | 18 | 21 | 61 |
| Travel | 9 | 21 | 15 | 7 | 52 |
| Food | 11 | 26 | 26 | 25 | 88 |
| Celebrity News | 8 | 10 | 5 | 9 | 32 |
| Cars | 16 | 13 | 9 | 16 | 54 |
| Crafts | 17 | 30 | 16 | 27 | 90 |
| Sports | 11 | 25 | 15 | 20 | 71 |
| Movies & TV | 10 | 11 | 6 | 16 | 43 |
| Games | 12 | 19 | 7 | 31 | 69 |
| Total | 151 | 278 | 281 | 292 | 1002 |

*Games, Movies & TV, Technology & Gadgets, Cars, and Celebrity News.*

**Note Types**   We analyzed UGC notes in the social platform style, which are marked by internet cultural symbols like buzzwords, slang, memes, and Chinese-specific neologisms, along with distinctive linguistic patterns. These include the "planting grass" discourse, used to share personal experiences with products or travel, or other highly expressive, often exaggerated emotional language.

Based on these features, we categorize the notes into four types:

1. **Informal General UGC Notes (General)**: Lacking prominent internet-specific symbols or distinctive expressive styles; resemble conventional casual writing.

2. **Notes Featuring Special Linguistic Style Expressions (Expres)**: Exhibit unique rhetorical or discursive strategies (e.g., "planting grass", exaggerated affective fram-

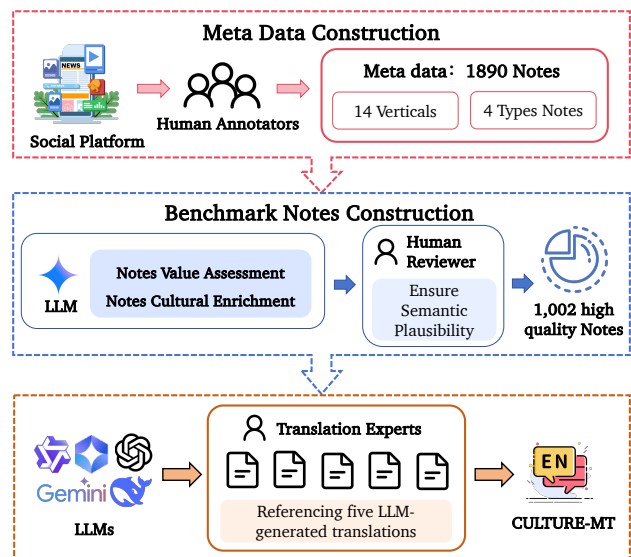

*Figure 4.* The pipeline of CULTURE-MT construction and annotation.

ing), with limited use of internet cultural symbols.

3. **Notes Rich in Internet Cultural Symbols (Symbol)**: Contain a high density of culturally grounded online expressions (e.g., trending phrases, memes, platform-specific jargon), but without marked linguistic stylistic idiosyncrasies.

4. **Hybrid Notes (Hybrid)**: Combine both rich internet cultural symbols and distinctive language expression styles.

### 3.2. Data Construction

Our data construction follows a human–LLM collaborative paradigm. As illustrated in Figure 4, human annotators first craft metadata reflecting authentic user scenarios, which is then rewritten and expanded by LLMs to produce the final benchmark data, along with approximately 100K augmented instances for training the Judger and translation models. The data augmentation pipeline is detailed in Appendix D.

**Meta Data**  We analyze the expression patterns of social platform users and manually construct data for four note types in each vertical, with 25, 40, 40, and 30 notes per type, totaling 1,890 notes. We also add topic tags to align with real UGC style. However, since topic tags on social platforms are linked to popularity metrics, translating them alongside notes may lead to multiple translations for the same tag, complicating real-world matching. To address this, we use placeholders for topic tags, such as "#占位0##占位1#" (which translates to "#Placeholder 0##Placeholder 1#" in English).

**Benchmark Data**  Building upon the human-curated metadata, we employ an LLM to perform two key operations: (1) *note value assessment* and (2) *cultural enrichment*. In the value assessment phase, the model filters notes based on two criteria: (i) the content must be translatable without reliance on images or videos, and (ii) the content should be valuable for non-Chinese audiences. This filtering yields 1,002 high-quality metadata entries. Next, for all non-General categories, we apply cultural enrichment—intentionally incorporating culturally loaded elements or accentuating linguistic style where appropriate. The enriched data undergoes a second round of human review to ensure semantic plausibility. The resulting 1,002 Chinese UGC notes constitute our benchmark dataset (dataset statistics are shown in Table 1). The prompts and LLM used in both stages are provided in the Appendix C.

**Filtering Analysis and Scope of CULTURE-MT**  We analyze the value-assessment filter to make its scope explicit. The retention rates are 43.1% for General, 49.6% for Express, 50.2% for Symbol, and 69.5% for Hybrid notes. Filtering removes more General notes, which are typically culturally sparse; among removed samples, 56.4% require multimodal context and 43.6% are short, low-content posts. The average token length increases from 71.17 to 104.11 after filtering, suggesting that many removed cases are difficult to evaluate from text alone, rather than merely culturally challenging. The final benchmark still contains 57.2% Symbol/Hybrid samples (573/1002 instances), and all 1,002 benchmark instances are manually inspected after enrichment. We therefore frame CULTURE-MT as a benchmark for text-evaluable cultural translation, while context-heavy, multimodal, or extremely covert cultural cases remain important future extensions.

**Human–LLM Collaborative Translation**  We utilized a collaborative annotation strategy combining LLMs and human experts. First, we select five open-source LLMs with strong Chinese–English capabilities for social media translation and prompt them to translate the 1,002 Notes into English. The Prompt (as shown in Figure 11) emphasizes adherence to native English expression norms while preserving the original cultural nuances. Human translators then synthesize these five machine-generated translations to produce a final, refined version. The synthesis stage is constrained: human translators choose the strongest machine-generated candidate as a base, then edit it to correct semantic errors, recover omitted cultural intent, and improve target-language naturalness, rather than freely rewriting the source content.

### 3.3. Automatic Evaluator Framework

We introduce an evaluation framework for cultural effectiveness in translation and a scored dataset to train an automatic

evaluator, JUDGER. Following MetricAlign (Zhang et al., 2025), we construct a validation set by sampling benchmark instances, generating translations from diverse open- and closed-source models, and collecting human annotations. We evaluate JUDGER on this set using Accuracy and Cohen's $\kappa$ to assess its agreement with human judgments.

**Definition of Cultural Effectiveness**  The primary goal of translating user-generated content on social platforms is to enable cross-lingual users to participate in the same community. While basic comprehensibility is necessary, true engagement hinges on whether the translation successfully conveys culturally embedded expressions that resonate with target-language users. We define this translation as **culturally effective** if it enables non-Chinese readers to correctly interpret the original intent and experience a comparable emotional or contextual response. For example, consider the Chinese colloquial expression: 这事儿吧，说破了就没意思了. A literal translation—"If we say it explicitly, it won't be interesting anymore"—misrepresents the pragmatic intent, as the expression relies on a shared cultural norm of implicit understanding rather than narrative suspense. A culturally effective translation would instead make the implicit social meaning explicit, such as: "It's one of those things better left unsaid—spelling it out would ruin the moment."

**Evaluation Guidelines**  Translation experts developed a scoring rubric (see Table 9) centered on two dimensions: *expressive accuracy* and *cultural adaptability*.

Expressive accuracy covers semantic fidelity and emotional tone, and, specifically in Chinese social media, the correct interpretation of *units/measures* and *proper nouns*. For example, units are often omitted (e.g., "160/90 Day1" implicitly refers to height/weight in cm/kg). Likewise, proper nouns should follow established translations: "Zhen Huan Biography." is conventionally better rendered as "Empresses in the Palace".

Cultural adaptability requires not only proper handling of culture-specific words or expressions but also alignment with target-language discourse norms, such that the translation reads naturally to native speakers. For example, 氛围感拉满 is awkwardly rendered as "The atmosphere feeling is pulled full" whereas a natural English equivalent is "The vibes are immaculate". Similarly, address forms require cultural interpretation: 刘亦菲们 does not denote individuals named Liu Yifei, but refers to stylish or aesthetically refined women, and can be translated as "style queens" or "gorgeous people".

Based on these criteria, experts assign scores on a 0–3 scale: 0 denotes severe loss of meaning or cultural intent, 1 denotes partially understandable but culturally ineffective transla-

*Table 2.* Models used in Judger evaluation datasets construction.

| Category | Models |
|---|---|
| Closed-Source | Gemini3 Pro High, Gemini3 Pro Low, GPT-5 |
| Open-Source | DeepSeek V3.2, Qwen3-235B-A22B, GLM-4.6 Qwen3-32B, Qwen3-8B, Qwen3-4B, Qwen3-1.7B, Qwen3-0.6B |

*Table 3.* Accuracy (Acc) and Cohen's $\kappa$ of Our Judger and Base LLMs Compared with Human Annotations. P: Precision, R: Recall, and F: F1-score.

| Model | ACC | Cohen's $\kappa$ | P-01 | R-01 | F1-01 | P-23 | R-23 | F1-23 |
|---|---|---|---|---|---|---|---|---|
| Qwen3-235B-A22B | 75.32% | 0.5064 | 81.93% | 64.97% | 72.47% | 70.98% | 85.67% | 77.63% |
| Qwen3-32B | 78.05% | 0.5610 | 95.31% | 59.01% | 72.89% | 70.32% | 97.09% | 81.56% |
| JUDGER | **86.03%** | **0.7205** | 84.25% | 88.66% | 86.40% | 88.00% | 83.38% | 85.63% |

tion, 2 denotes generally effective translation with remaining stylistic or cultural weaknesses, and 3 denotes highly effective translation. However, due to the paragraph-level nature of UGC notes—where quality may vary across sentences—the boundary between scores 0–1 and 2–3 is often ambiguous. Consequently, we treat the task as binary: scores 0–1 indicate *ineffective* cultural transmission, while 2–3 indicate *effective* transmission. There are case studies with different scores that can be found in Appendix J.

**Training the Judger**  Following the above guidelines, we first sample 3,000 instances from the UGC data constructed in Section 3.2 and obtain expert annotations. These 3,000 expert-labeled samples are drawn from the augmented UGC pool rather than from the 1,890 metadata records or the 1,002 benchmark evaluation notes, ensuring separation between evaluation and training sources. The corresponding translations are generated by multiple models listed in Table 2, rather than by Gemini alone. We then use Gemini-3 (Google, 2025) to automatically annotate an additional 40,000 samples using the prompt shown in Figure 13. From the combined dataset, we perform score-balanced sampling to select 30,000 instances for training. The score-balanced sampling removes many high-scoring examples from stronger generators and mitigates preference toward a single model style. We fine-tune Qwen3-32B using supervised fine-tuning (SFT) to obtain the JUDGER model.

**Judger Evaluation**  We construct a test set by sampling six notes per domain (1+2+2+1 across four note types) from the benchmark, yielding 84 source notes in total. For each note, we generate translations using 11 diverse models (Table 2) and scored by human experts. Two experts independently annotate all evaluation samples; disagreements are resolved through third-party adjudication. As summarized in Table 4, the experts reach 72% agreement on the four-class score and 93% agreement after binary grouping, and Gemini annotations achieve 88.00% accuracy and Cohen's $\kappa$ of 0.76 against expert gold labels. We obtain 688 translation instances to evaluate our JUDGER

*Table 4.* Reliability results for cultural-effectiveness annotation.

| Item | Metric | Value |
|---|---|---|
| Expert–expert agreement | Four-class Acc. | 72% |
| Expert–expert agreement | Binary Acc. | 93% |
| Gemini annotation vs. expert gold | Acc. | 88.00% |
| Gemini annotation vs. expert gold | Cohen's $\kappa$ | 0.76 |

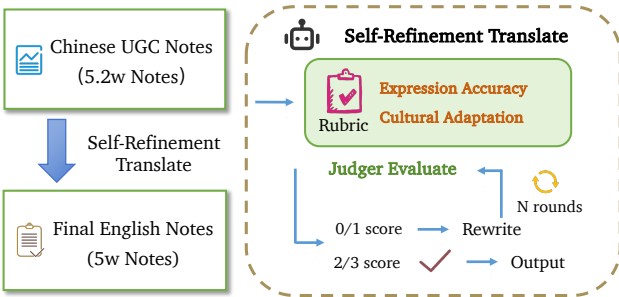

*Figure 5.* Cultural effectiveness–based self-refinement pipeline for constructing UGC translation training data.

*Table 5.* Results on cultural effectiveness and the detail scores distribution. Ineff. denotes the proportion of samples with cultural effectiveness scores 0-1 (lower is better). Eff. proportion of samples scores 2-3. AVG. is the average score in all 1002 samples.

| Models | Cultural Effectiveness | | Score Distribution | | | | AVG. |
|---|---|---|---|---|---|---|---|
| | Ineff. ↓ | Eff. | 0↓ | 1↓ | 2 | 3 | |
| Close-source LLMs | | | | | | | |
| Gemini 3 | **9.20%** | **90.80%** | 0.10% | 9.10% | 52.50% | 38.30% | **2.29** |
| GPT-5 | 9.38% | 90.62% | 1.00% | 8.38% | 53.89% | 36.73% | 2.26 |
| Open-source Base LLMs (>100B) | | | | | | | |
| Deepseek V3.2 | 16.57% | 83.43% | 3.09% | 13.47% | 58.08% | 25.35% | **2.06** |
| GLM 4.6v | **16.07%** | **83.93%** | 3.79% | 12.28% | 58.48% | 25.45% | **2.06** |
| Qwen3-235B-A22B | 30.24% | 69.76% | 12.67% | 17.56% | 52.69% | 17.07% | 1.74 |
| Qwen3 Series LLMs | | | | | | | |
| Qwen3-32B | **30.74%** | **69.26%** | 5.79% | 24.95% | 59.38% | 9.88% | **1.73** |
| Qwen3-14B | 41.32% | 58.68% | 13.67% | 27.64% | 51.00% | 7.68% | 1.53 |
| Qwen3-8B | 49.60% | 50.40% | 15.17% | 34.43% | 46.31% | 4.09% | 1.39 |
| Qwen3-4B | 60.98% | 39.02% | 21.66% | 39.32% | 36.13% | 2.89% | 1.20 |
| Qwen3-1.7B | 83.53% | 16.47% | 47.21% | 36.33% | 15.97% | 0.50% | 0.7 |
| Qwen3-0.6B | 95.41% | 4.59% | 74.65% | 20.76% | 4.39% | 0.20% | 0.3 |
| Open-source Translation LLMs (7B) | | | | | | | |
| Seed-X-PPO | 31.04% | 68.96% | 3.89% | 27.15% | 62.77% | 6.19% | **1.71** |
| Seed-X-Instruct | 49.30% | 50.70% | 4.79% | 44.51% | 48.90% | 1.80% | 1.48 |
| UGC Translation Baselines | | | | | | | |
| Ours-32B | 13.97% | 86.03% | 1.40% | 12.57% | 58.18% | 27.84% | 2.12 |
| Ours-8B | 14.47% | 85.53% | 0.50% | 13.97% | 57.29% | 28.24% | **2.13** |

after balanced sampling, and the results are reported in Table 3. JUDGER achieves an overall accuracy of 86.03% and a Cohen's Kappa of 0.7205, indicating substantial agreement with human judgments. Notably, recall score reaches 88.66% for ineffective cases and 83.38% for effective cases, suggesting a slightly conservative bias toward cultural effectiveness.

## 4. UGC Translate Baselines

To establish strong UGC translation baselines for CULTURE-MT, we construct a training corpus with culturally effective translation and fine-tune two variants of the Qwen3 model to produce specialized UGC translation models, serving as baselines for CULTURE-MT.

**Cultural Effectiveness UGC Training Data** Specifically, we sample 52K augmented notes (as described in Section 3.2) and annotate their translation outputs via a Cultural Effectiveness–based self-refinement procedure, yielding 50K UGC translation training instances. As shown in Figure 5, for each of the 52K augmented notes, we first generate an initial English translation using Gemini-3 (Google, 2025). The resulting translations are then evaluated by the JUDGER (Section 3.3) to assess cultural effectiveness. Samples receiving low scores (0–1) are iteratively rewritten by Gemini-3 guided by the evaluation feedback, for up to $n$ iterations. Translations that fail to reach the effectiveness threshold after $n$ rounds are discarded. We set $n = 3$ to obtain high-quality training data. This iterative *generate–evaluate–rewrite* loop yields a high-quality and culturally consistent UGC translation corpus, as illustrated in Figure 5. This translation-model training path is separated from

the benchmark evaluation labels and from the multi-model translation pool used to train JUDGER.

**Training Setting** We fine-tune Qwen3-8B and Qwen3-32B on the resulting ∼50K UGC translation instances. The training prompt is shown in Figure 11. For both models, we use a batch size of 512 and a learning rate of $1 \times 10^{-5}$, and train for one epoch with a warmup ratio of 0.05. Full-parameter fine-tuning is performed on 2×H800 GPUs for Qwen3-8B and 4×H800 GPUs for Qwen3-32B, respectively, using DeepSpeed with ZeRO Stage-3 optimization.

## 5. Experiment

### 5.1. Baseline LLM Performance on CULTURE-MT

**Baselines** Using CULTURE-MT and our JUDGER, we comprehensively evaluate LLMs with different architectures and scales on UGC note translation. The baselines include (1) **closed-source LLMs**: GPT-5 and Gemini 3, both with strong general-purpose multilingual capabilities; (2) **large-scale open-source LLMs**: Deepseek V3.2, GLM 4.6V and Qwen3-235B-A22B, whose parameter sizes exceed 100B and which perform strongly in Chinese and English; (3) several **Qwen3-series open-source models**: 0.6B, 1.7B, 4B, 8B, 14B and 32B; (4) **open-source translation LLMs**: Seed-X-Instruct and Seed-X-PPO (Cheng et al., 2025); and (5) **our UGC translation baselines** described in Section 4. The prompts used by each LLM to generate translations are shown in Figure 11.

**Performance** Table 5 reports the cultural effectiveness of CULTURE-MT benchmark on different models. Overall, larger models exhibit a substantially higher proportion of cultural effective translations. Closed-source models, Gem-

*Table 6.* Ablation of the JUDGER-guided rewrite loop on CULTURE-MT.

| Model | Ineff. ↓ | Eff. ↑ | AVG. |
|---|---|---|---|
| DeepSeek V3.2 | 16.57% | 83.43% | 2.06 |
| Ours-32B | 13.97% | 86.03% | 2.12 |
| Ours-8B | 14.47% | 85.53% | 2.13 |
| Ours-32B w/o JUDGER loop | 16.87% | 83.13% | 2.07 |
| Ours-8B w/o JUDGER loop | 16.17% | 83.83% | 2.10 |

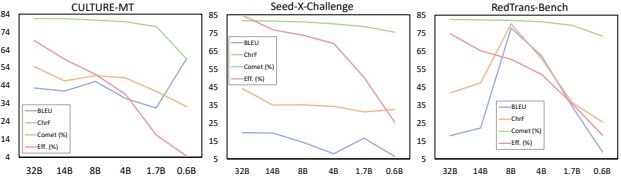

*Figure 6.* Performance of the Qwen3 model family at different model scales on three datasets, evaluated using BLEU, ChrF, COMET, and Eff., where Eff. denotes the proportion of samples exhibiting cultural effectiveness as defined in this work.

ini 3 and GPT-5, achieve over 90% of samples in the 2–3 score range with 2.29 and 2.26 average score, demonstrating robust cross-cultural understanding and emotion resonance. Among open-source base models, Deepseek V3.2 and GLM 4.6v significantly outperform smaller models, yet still trail behind closed-source systems. The Qwen3 series reveals a clear degradation trend as model size decreases: lower-capacity models show a sharp increase in 0–1 score samples and an almost complete disappearance of score-3 outputs. **This suggests that cultural effectiveness is particularly sensitive to the size of base LLMs, especially for capturing stylistic and sociopragmatic nuances in UGC.**

For specialized translation models, Seed-X-PPO attains a high proportion of score-2 samples but produces relatively few score-3 translations with only 1.71 average score, indicating that high general translation quality does not necessarily bring strong cultural resonance. Seed-X-Instruct further exhibits a high share of low-score samples, highlighting the difficulty of generalizing translation-oriented training to social media contexts. Notably, our approach based on Qwen3-8B substantially increases the proportion of score-3 samples to 28.24%, outperforming the corresponding base model (4.09%) by a large margin. **This improvement demonstrates that incorporating UGC-aware cultural modeling can effectively enhance cultural effectiveness beyond what is achievable through scale or translation-centric training alone.**

Table 6 isolates the contribution of the JUDGER-guided rewrite loop. Removing the loop increases ineffective translations from 13.97% to 16.87% for Ours-32B and from 14.47% to 16.17% for Ours-8B, showing that in-domain data provides a strong foundation while cultural-effectiveness-

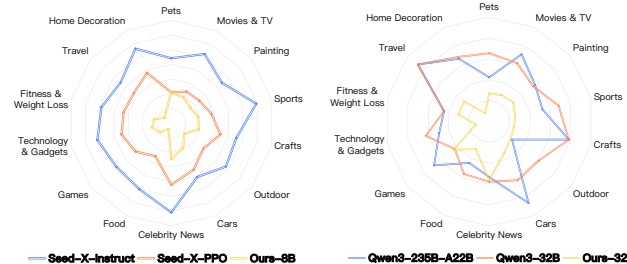

*Figure 7.* The domain-wise cultural ineffective share.

guided refinement brings additional targeted gains.

We evaluate automatic metrics (BLEU, ChrF, COMET) against cultural effectiveness. As illustrated in the left panel of Figure 6, these automatic metrics exhibit only minor variation across model scales and fail to capture the pronounced differences among models—especially in 32B, 14B, and 8B variants—in terms of cultural adaptation. In stark contrast, cultural effectiveness demonstrates a clear and consistent scaling trend. This divergence underscores that conventional automatic metrics are largely blind to culturally grounded translation errors, whereas **cultural effectiveness offers a more discriminative and meaningful signal for evaluating UGC translation quality.** There are case studies of JUDGER output in Figure 15.

### 5.2. Analysis

**Domain-wise analysis** The results in Figure 7 reveal substantial domain- and model-level variation in cultural ineffectiveness. Seed-X-Instruct exhibits consistently high Ineff. rates across domains, indicating that instruction tuning alone is insufficient for culturally grounded UGC translation. Seed-X-PPO reduces ineffective cases, confirming the benefit of preference optimization, yet remains vulnerable to domain shifts. In contrast, Ours-8B consistently achieves the lowest ineffectiveness rates, with particularly pronounced gains in culture-intensive domains, such as Food, Games, Sports, and Celebrity News, where translations depend heavily on idiomatic or community-specific expressions. The performance gap narrows in more content-neutral domains (e.g., Movies & TV and Painting), suggesting that cultural adaptation matters most when meaning is conveyed implicitly by cultural-load symbols. **Overall, these findings demonstrate that cultural effectiveness–oriented training not only elevates average translation quality but also enhances robustness across diverse UGC vertical.**

**Note types-wise analysis** Figure 8 analyzes the cultural ineffective share across different Note types. Across most models, the ineffective rate increases from General to Express Notes, and reaches its highest level on Symbol and Hybrid Notes. This trend suggests that cultural effectiveness

*Table 7.* Evaluation results across CULTURE-MT, Seed-X-Challenge, RedTrans, and FLORES with both automatic metrics (BLEU, ChrF, COMET(%)) and Ineff.(%).

| Models | CULTURE-MT | | | | Seed-X | | | | RedTrans | | | | FLORES | | |
|---|---|---|---|---|---|---|---|---|---|---|---|---|---|---|---|
| | BLEU | ChrF | COMET | Ineff. ↓ | BLEU | ChrF | COMET | Ineff. ↓ | BLEU | ChrF | COMET | Ineff. ↓ | BLEU | ChrF | COMET |
| **Closed-source LLMs** | | | | | | | | | | | | | | | |
| Gemini 3 | 35.77 | 52.64 | 85.04 | 9.20 | 11.84 | 39.37 | 80.96 | 6.60 | 18.80 | 47.30 | 80.93 | 17.34 | 51.47 | 49.79 | 87.92 |
| GPT-5 | 44.49 | 51.33 | 83.76 | 9.38 | 8.53 | 31.24 | 81.58 | 6.09 | 13.67 | 48.16 | 81.25 | 18.09 | 28.43 | 43.99 | 80.36 |
| **Open-source LLMs** | | | | | | | | | | | | | | | |
| Deepseek V3.2 | 39.52 | 47.86 | 82.72 | 16.57 | 20.89 | 44.28 | 82.20 | 11.17 | 19.34 | 33.80 | 81.01 | 23.45 | 28.64 | 50.35 | 87.91 |
| GLM 4.6v | 35.18 | 53.54 | 83.12 | 16.07 | 12.14 | 40.24 | 82.20 | 12.18 | 18.14 | 34.44 | 81.23 | 21.73 | 58.35 | 58.34 | 88.13 |
| Qwen3-235B-A22B | 39.29 | 51.32 | 81.79 | 30.24 | 14.01 | 45.05 | 82.13 | 16.75 | 27.82 | 42.70 | 81.40 | 29.87 | 27.55 | 45.62 | 87.76 |
| **Qwen3 Series LLMs** | | | | | | | | | | | | | | | |
| Qwen3-32B | 42.68 | 54.63 | 81.54 | 30.74 | 19.47 | 44.01 | 81.77 | 15.23 | 17.89 | 41.73 | 82.62 | 25.37 | 28.49 | 43.03 | 87.79 |
| Qwen3-14B | 40.93 | 46.62 | 81.49 | 41.32 | 19.31 | 34.91 | 81.45 | 23.35 | 22.14 | 47.27 | 82.39 | 34.80 | 26.55 | 49.51 | 87.67 |
| Qwen3-8B | 46.33 | 49.66 | 80.62 | 49.60 | 14.15 | 34.95 | 81.03 | 26.40 | 77.87 | 80.13 | 82.10 | 39.61 | 58.06 | 60.38 | 87.38 |
| Qwen3-4B | 36.77 | 48.30 | 79.68 | 60.98 | 7.84 | 34.15 | 79.98 | 30.96 | 62.33 | 60.68 | 81.31 | 48.07 | 25.17 | 42.56 | 87.03 |
| Qwen3-1.7B | 31.47 | 40.79 | 77.05 | 83.53 | 16.46 | 31.06 | 78.48 | 49.75 | 34.23 | 36.43 | 79.39 | 64.24 | 49.54 | 47.52 | 86.04 |
| Qwen3-0.6B | 59.20 | 32.25 | 58.98 | 95.41 | 6.36 | 32.43 | 75.35 | 74.62 | 8.82 | 25.47 | 73.29 | 81.80 | 21.54 | 37.50 | 82.58 |
| **Open-source Translation LLMs (7B)** | | | | | | | | | | | | | | | |
| Seed-X-PPO | 34.84 | 47.08 | 80.18 | 31.04 | 14.12 | 41.30 | 81.28 | 19.29 | 53.39 | 67.24 | 83.70 | 27.96 | 52.54 | 55.18 | 87.60 |
| Seed-X-Instruct | 35.64 | 47.33 | 81.60 | 49.30 | 10.93 | 35.76 | 79.52 | 29.44 | 34.57 | 56.43 | 83.38 | 33.69 | 28.36 | 52.44 | 87.65 |
| **Ours Models** | | | | | | | | | | | | | | | |
| Ours-32B | 38.91 | 44.18 | 81.68 | 13.97 | 8.47 | 41.82 | 77.44 | 25.38 | 15.51 | 33.03 | 77.74 | 19.00 | 56.75 | 55.77 | 87.44 |
| Ours-8B | 43.21 | 41.41 | 81.69 | 15.77 | 6.48 | 33.92 | 77.77 | 24.87 | 18.33 | 42.41 | 78.24 | 21.62 | 15.92 | 37.88 | 84.47 |

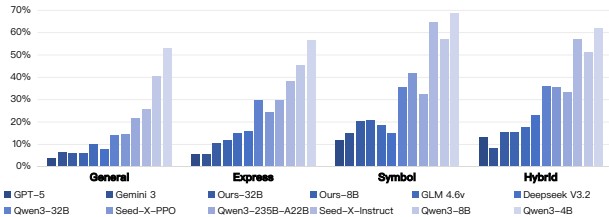

*Figure 8.* Cultural ineffective share across different note types.

degrades as linguistic expressiveness and cultural symbol density increase. In particular, Symbol and Hybrid Notes pose the greatest challenge, as they rely heavily on implicit meanings, symbolic references, and cultural context.

### 5.3. Multi-Dataset and Multi-Metric Evaluation

Table 7 reports results on CULTURE-MT, Seed-X, Red-Trans, and FLORES using both automatic metrics (BLEU, ChrF, COMET) and cultural effectiveness by JUDGER. While standard automatic metrics exhibit limited discriminability, particularly on UGC-oriented benchmarks, our Ineff. metric reveals markedly sharper performance separation, as shown in Figure 6. Crucially, it highlights the advantage of explicit cultural alignment: our models (Ours-8B, Ours-32B) achieve the low Ineff. scores across CULTURE-MT, Seed-X-Challenge, and RedTrans, even when their automatic scores are comparable to or below those of general-purpose LLMs. This demonstrates that cultural effectiveness captures a dimension of UGC translation quality orthogonal to fluency and adequacy, which conventional metrics fail to reflect.

Moreover, the consistent discriminative power of Ineff. across diverse benchmarks, including Seed-X-Challenge and RedTrans, indicates that the JUDGER is robust and generalizable beyond CULTURE-MT. Notably, while our UGC-specialized 8B model shows a modest decline in BLEU/COMET on FLORES, the degradation is small. This suggests that cultural-effectiveness–driven training yields a favorable trade-off: substantial gains on UGC tasks with only marginal costs on broad-domain performance.

## 6. Conclusion

In this work, we move beyond literal translation and study cultural effectiveness as a central requirement for social media UGC translation. We introduce CULTURE-MT, a benchmark designed to capture culturally grounded and expressive content, and propose JUDGER, an automatic evaluator that assesses translation quality beyond surface-level semantic accuracy. Our results show that cultural effectiveness varies systematically across domains and note types, and that standard automatic metrics are insufficient to reflect these differences. By explicitly modeling cultural-loaded terms and linguistic style, cultural effectiveness–oriented evaluation and training substantially improve the robustness of UGC translation, while only marginally affecting general translation performance. Overall, this work highlights the necessity of moving beyond literal correctness toward culturally effective translation for real-world social media applications. We contribute a challenging benchmark tailored to this scenario, and our proposed cultural effectiveness metric provides a principled and actionable signal to guide the development and optimization of translation models for culturally aware UGC generation.

## Acknowledgements

This work is supported by the National Natural Science Foundation of China (No. 62376245), National

Key Research and Development Project of China (No. 2018AAA0101900), MOE Engineering Research Center of Digital Library, and the industry-university collaboration project between Xiaohongshu and Zhejiang University.

## Impact Statement

We introduce **CULTURE-MT**, a benchmark for evaluating the cultural effectiveness of social media translation systems, with a focus on culture-loaded symbols and linguistic styles in user-generated content (UGC). It provides a structured framework to assess cultural adaptation in cross-lingual UGC translation, complementing existing NLP benchmarks by emphasizing cultural resonance alongside linguistic accuracy. In this work, culturally effective translation systems can enhance cross-cultural communication in applications such as multilingual customer service, social media interaction, content moderation, and digital marketing. By preserving cultural nuance, these systems may foster more authentic and inclusive global dialogue.

This study adheres to established ethical guidelines. Benchmark datasets are publicly available and contain no personally identifiable information. No human participants were directly involved in experiments. Human annotation, where used, was conducted by trained annotators under fair labor practices.

While we aim to promote culturally aware translation without generating harmful content, limitations remain. Augmentation data generated by LLMs may include unexpected or sensitive expressions, and the JUDGER component may not fully capture cultural interpretations across all languages or communities. In addition, our current benchmark prioritizes self-contained textual notes; comments, very short posts, video notes, and cases requiring user/thread-level or multimodal context may be underrepresented. Future work should extend CULTURE-MT toward contextual and multimodal UGC evaluation and continually refresh emerging cultural phenomena. Evaluation results should therefore be interpreted with caution. We encourage future research to investigate risks associated with culturally adaptive systems, including methods to detect and mitigate harmful content arising from misinterpretation or malicious use. Rather than focusing solely on detecting machine-generated text, efforts should prioritize preventing the propagation of culturally harmful outputs.

The research artifacts, including benchmark datasets, evaluation tools, and models—are released exclusively for research and educational purposes. The authors assume no liability for downstream uses and urge the community to critically examine biases, limitations, and responsible deployment practices in culturally aware translation systems.

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

## A. Open-Source Resources and Leaderboard

To support reproducibility and facilitate future research, we have released the CULTURE-MT benchmark on Hugging Face: `https://huggingface.co/datasets/Wulinjuan/CULTURE-MT`.

In addition, we have built an online leaderboard for CULTURE-MT: `https://huggingface.co/spaces/Wulinjuan/CULTURE-MT`.

The leaderboard allows researchers and practitioners to submit translation results and evaluate them using our trained JUDGER model. By providing both the benchmark data and an automatic evaluation interface, we aim to make CULTURE-MT easier to use, compare, and extend. We hope these resources can support more systematic evaluation of cultural effectiveness in social media UGC translation and encourage future work on culturally aware machine translation.

## B. Prompts for Benchmark Construction

We utilized Gemini-3 (Google, 2025) to construct Notes for CULTURE-MT. We set prompts for Note value assessment (Figure 9) and cultural enrichment (Figure 10).

You are a social media expert. Given a piece of content, determine whether it is suitable for inclusion in a UGC translation benchmark.

Please evaluate the content according to the following two criteria:

1. **Translation Feasibility**:
   Determine whether the content can be accurately translated without relying on additional visual information, such as images or videos.

2. **Cross-lingual Value**:
   Determine whether the content is worth recommending to English-speaking or other non-Chinese users, i.e., whether it is relevant, informative, or appealing to audiences beyond the Chinese-speaking community.

Only if both criteria are satisfied, return a positive decision. Your output must strictly follow the format below:

```
Reason: xxx
Final Decision: Yes/No
```

Content to evaluate:
{content}

*Figure 9.* The prompt for Note value assessment.

## C. Prompt for CULTURE-MT Translation Generation

Figure 11 shows the prompt for our CULTURE-MT benchmark to generate translation from Chinese to English. For titles and content in a note, we use structural tags such as "`<title></title>`" for titles and "`<content></content>`" for content.

## D. UGC Data Augmentation

We expand our 1890 metadata to about 100,000 Notes using a two-step data generation pipeline based on Gemini-3 (Google, 2025).

**1. Topic suggestion.** For each (Domain, Note) pair from metadata, we prompt an LLM with real examples and ask: *"What topics would users in this domain want to share or read about?"* The model returns a short list of plausible, user-motivated topics.

**2. Note generation.** For each suggested topic, we prompt the LLM again to write a full note, conditioned on the domain, note type, and 1–2 metadata examples as style references.

We multi-sample with temperature = 1 and top-p = 1, and ensure that all generated notes are sufficiently dissimilar from the original 1,890 metadata instances to prevent overlap with the test set. We further remove near-duplicates among generated

You are an expert in social media culture research. You will be given a user-generated Note related to the {topic} topic with {type} type, including a title and content. Please enhance the Note according to its assigned type, following the rules below:

1. **Express Notes**
• Determine whether the note exhibits a strong expressive or emotional writing style, such as recommendation-style ("planting grass"), review-style, checklist-style, emotional resonance, or interactive prompting.
• If the expressive characteristics are not sufficiently clear, moderately enhance the stylistic features.
• Do not introduce slang, buzzwords, or other culture-loaded terms for this type.

2. **Symbol Notes**
If the note contains few culture-loaded elements, appropriately add one or two of the following:
    • Internet slang, meme terms, or buzzwords associated with Chinese online culture
    • Culture-specific content such as Chinese idioms, sayings, or classical expressions

3. **Hybrid Notes**
Assess both linguistic style and the presence of culture-loaded terms. Enhance the note so that it exhibits **clear expressive stylistic features** while also containing **rich culture-loaded symbols**.

**General Requirements:**
• Do not make extensive modifications.
• Preserve semantic coherence and accuracy.
• Ensure the rewritten text conforms to natural Chinese UGC writing conventions.

**Input:**
Title: {title}
Content: {content}

**Output:**
The rewritten output should follow the same format as the input.

*Figure 10.* The prompt for Note cultural enrichment.

notes (sentence embedding cosine similarity >0.75) and spot-check <5% of outputs for realism. To examine fine-grained diversity beyond the coarse 14-domain × 4-note-type taxonomy, we cluster generated notes within each vertical by sentence embeddings and manually inspect representative clusters; examples are reported in Appendix E.

## E. Fine-Grained Topic Clustering Evidence

Table 8 presents representative within-taxonomy clusters from the augmented data. The clusters show that the augmented corpus covers not only broad domains but also diverse subtopics, user intents, and expression scenarios within each domain.

*Table 8.* Representative fine-grained topic clusters within selected taxonomies.

| Taxonomy | Representative clusters |
|---|---|
| Sports | running; marathon logs; badminton partner-seeking; swimming practice; tennis; football league discussion; table-tennis fandom; NBA/basketball discussion; boxing/combat sports; billiards/golf |
| Outdoor | skiing; camping; hiking; mountaineering; cycling; diving; surfing; rock climbing; fishing; beach-combing; drifting; paddleboarding |
| Home & Decoration | renovation/design; whole-house customization; kitchen/bathroom organization; furniture and bedding; curtains/windows; home appliances; fragrance/flowers/tea ware; waterproofing; formaldehyde removal |
| Pets | cats; dogs; adoption/rescue; pet services; pet food; pet health products; birds; rabbits; hamsters; turtles; ornamental fish |

## F. Rubric for Cultural Effectiveness for UGC Translation

Translation experts developed a scoring rubric (see Table 9) centered on two dimensions: *expressive accuracy* and *cultural adaptability*.

You are a multilingual translation expert specialized in translating Chinese user-generated social media content into English. Please translate the following [CONTENT] from Chinese into English.

Important instructions:

1. Placeholders in the form of #占位x# must not be translated. Preserve each placeholder exactly and keep it in the corresponding position in the translated text.

2. Output only the translated result. Do not include any additional content, notes, or explanations.

3. Preserve all structural tags such as <title></title>, <content></content>, etc., and output the translation using the same format.

4. The source text is user-generated social media content. The translation should:
   - Faithfully preserve the original emotional tone (e.g., excitement, sarcasm, complaint, humor, irony).
   - Sound natural and idiomatic to native English speakers on social platforms.
   - Avoid overly formal, academic, or literal phrasing.

5. When the original text contains culture-specific expressions, slang, or implicit context that may not be immediately clear to English readers:
   - Translate them into an equivalent expression that conveys the same intent and emotion.
   - Do not add explanations or annotations.

6. Do not introduce information that is not present in the original text.
   Do not omit emotionally or pragmatically important details or structure.

The [CONTENT] to be translated is as follows:

{content}

Please provide the translated result:

*Figure 11.* The prompt for CULTURE-MT translation generation.

## G. Judger Training Flowchart

Figure 12 illustrates the score guidelines for cultural effectiveness, training data construction, and evaluation data construction of the JUDGER system.

## H. The results in Different Domains

We report the domain-wise cultural ineffective share results in 11 models with $\geq$ 4B, as shown in Table 10. Culture-intensive domains such as *Games* and *Celebrity News* exhibit the highest ineffective rates, likely due to dense slang, implicit references, and community-specific expressions, whereas more descriptive domains (e.g., *Travel* and *Home Decoration*) are relatively easier.

## I. The Prompt for Cultural Effectiveness Evaluation

The prompt used for LLM-based annotation during JUDGER training, as well as for evaluating cultural effectiveness with JUDGER, is shown in Figure 13 (Chinese version) and Figure 14 (Translation version).

## J. Case Study

### J.1. Judger-Guided Case Analysis

Figure 15 presents an example translated by Qwen-8B that is rated as culturally ineffective (score 0), along with the corresponding reasoning produced by the JUDGER. **The detailed comments produced by the JUDGER not only explain the evaluation outcome but also serve as supervision signals for guiding future correction and refinement of culturally ineffective translations.**

### J.2. Translation Cases with Different Cultural Effectiveness Score

Figure 16 presents translation examples for the same case, with scores ranging from 0 to 3, demonstrating varying degrees of cultural effectiveness. These scores reflect the high standards of CULTURE-MT's cultural effectiveness criterion (Eff.).

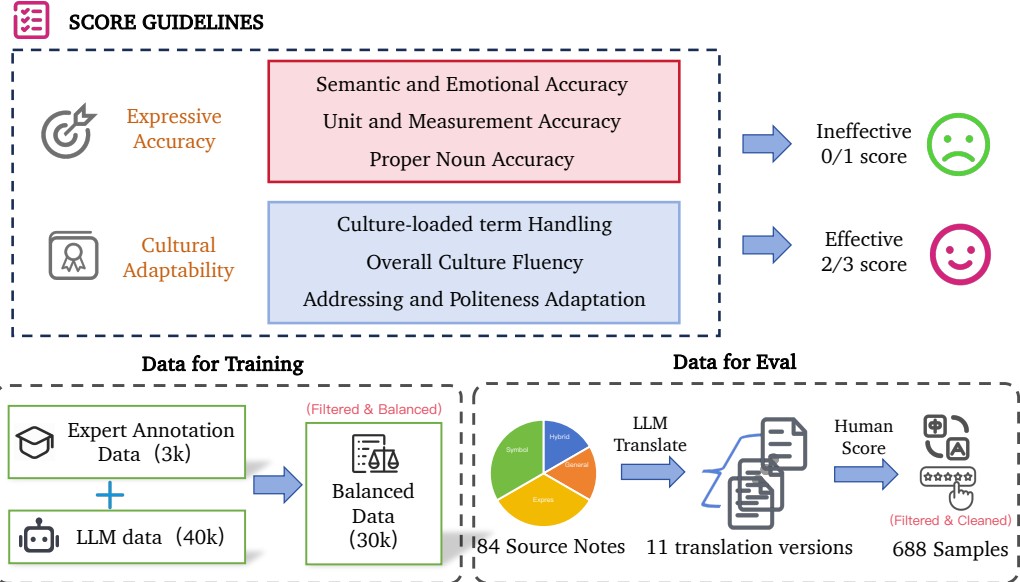

*Figure 12.* Key components of the JUDGER construction process, including score guidelines, training data construction, and evaluation data construction. The score guidelines assist both humans and models in evaluating the cultural effectiveness of translations.

The Ours-8B translation effectively conveys the humor and cultural intent of the original, while other translations struggle with emotional expression and cultural adaptation, resulting in lower effectiveness scores.

The 0-score translation (Qwen3-8B) is penalized for its awkward phrasing and lack of emotional depth, which makes it unclear and disjointed, failing to capture the original tone. The 1-score translation (Ours-32B) conveys the main idea but is criticized for its stiff phrasing and lack of fluency, making it feel less natural and emotionally resonant. The 2-score translation (Qwen3-32B) effectively communicates the core meaning but falls short in capturing the emotional nuances of the original, using clumsy expressions that hinder its natural flow and depth.

## K. Human Annotation and API Costs

The entire process of building our benchmark, training and testing the Judger model, and training the translation model involved outsourcing annotation and translation experts for manual construction and labeling, at a cost of $12,247.23. Additionally, API calls were made, with the primary expense being $943.65 for Gemini-3-pro, bringing the total to $13,190.88.

*Table 9.* Evaluation Criteria for Chinese–English Social Media Translation

| Dimension | Sub-dimension | Requirement | Key Points | Examples |
|---|---|---|---|---|
| **Expression Accuracy** | Semantic and Emotional Accuracy | Whether the translation is complete and accurately conveys both the literal meaning and implicit emotions of the source text (including sentiment, tone, and intent). | 1. No loss or distortion of factual information. 
 2. Emotional tone matches the original (e.g., excitement, sarcasm, frustration). 
 3. Correct handling of pragmatic inference and ambiguity caused by discourse focus shifts. | **Source:** "我真的会谢！" 

 **Reference:** "I'm literally done." (Correct emotional expression) 

 **Incorrect:** "I will really thank you." (Literal translation, wrong emotion) |
| | Unit and Measurement Accuracy | When culturally specific units are involved, whether conversions are accurate, clear, and whether implicit information in Chinese is properly supplemented when necessary. | 1. Correct numerical conversion. 
 2. Clear and unambiguous units. 
 3. Necessary contextual supplementation for omitted units in Chinese. | **Source:** "他180斤。" 

 **Reference:** "He weighs 180 *jin* (about 90 kilograms)." 

 **Incorrect:** "He weighs 180." (Unit missing) 

 **Incorrect:** "He weighs 180 pounds." (Wrong unit) |
| | Proper Noun Accuracy | For names of people, places, brands, and works, whether official or widely accepted translations are used. | 1. Use standardized translations (e.g., "北京" → "Beijing"). 
 2. When no official translation exists, adopt common transliteration or convention-based translations. 
 3. Maintain consistency throughout the text. | **Source:** "看了《甄传》。" 

 **Reference:** "Watched *Empresses in the Palace*." 

 **Incorrect:** "Watched *Zhen Huan Biography*." (Non-standard) |
| **Cultural Adaptation** | Culture-loaded Term Handling | Whether idioms, slang, and platform-specific expressions are appropriately interpreted, explained, or culturally adapted to ensure comprehension by target readers. | 1. Avoid destructive literal translation. 
 2. Prefer culturally equivalent expressions in the target language. 
 3. Use explanatory translation when necessary to integrate naturally into context. | **Source:** "这课太水了。" 

 **Reference:** "This course is basically filler." (Colloquial English) 

 **Incorrect:** "This course has too much water." (Literal translation) |
| | Overall Cultural Fluency | Whether the translation aligns with usage norms of English social media and reads like original content rather than a translation. | 1. Conforms to English social media style and lexical preferences. 
 2. Avoids Chinese-style English syntactic patterns. 
 3. Overall fluent, natural, and platform-native. | **Source:** "氛围感拉满。" 

 **Reference:** "The vibes are absolutely immaculate." 

 **Incorrect:** "The atmosphere feeling is pulled full." |
| | Addressing and Politeness Adaptation | Whether culturally specific forms of address and vocatives are adapted to fit social norms and politeness conventions of the target culture. | 1. Consider appropriate levels of familiarity and context. 
 2. Avoid awkwardness or unintended offense caused by literal address translation. 
 3. Seek functionally equivalent expressions in the target culture. | **Source:** "姐妹们看过来！" 

 **Reference:** "Hey guys, check this out!" 

 **Incorrect:** "Sisters, look here!" (Awkward, slogan-like) |

*Table 10.* Domain-wise **Ineffective Share** (**score 0–1**; lower is better, ↓).

| Domain | Seed-X-PPO | Seed-X-Instruct | Ours-8B | Qwen3-8B | Ours-32B | Qwen3-32B | Qwen3-4B | Qwen-235B | Gemini 3 | Deepseek V3.2 | GLM-4.6v | AVG. |
|---|---|---|---|---|---|---|---|---|---|---|---|---|
| Outdoor | 33.33% | 48.81% | 39.29% | 47.62% | 25.00% | 30.95% | 58.33% | 35.71% | 7.14% | 20.24% | 15.48% | 32.90% |
| Pets | 34.83% | 51.69% | 39.33% | 46.07% | 25.84% | 26.97% | 61.80% | 28.09% | 10.11% | 20.22% | 13.48% | 32.58% |
| Travel | 32.69% | 44.23% | 42.31% | 53.85% | 26.92% | 32.69% | 50.00% | 21.15% | 11.54% | 17.31% | 7.69% | 30.94% |
| Food | 23.86% | 52.27% | 48.86% | 48.86% | 29.55% | 34.09% | 67.05% | 26.14% | 4.55% | 15.91% | 21.59% | 33.88% |
| Games | 42.03% | 60.87% | 46.38% | 63.77% | 33.33% | 39.13% | 76.81% | 39.13% | 10.14% | 20.29% | 23.19% | 41.37% |
| Movies & TV | 20.93% | 44.19% | 48.84% | 34.88% | 32.56% | 30.23% | 60.47% | 13.95% | 6.98% | 18.60% | 20.93% | 30.23% |
| Home Decoration | 25.00% | 50.00% | 48.53% | 47.06% | 36.76% | 27.94% | 58.82% | 22.06% | 4.41% | 5.88% | 11.76% | 30.75% |
| Sports | 38.03% | 56.34% | 47.89% | 42.25% | 22.54% | 21.13% | 56.34% | 33.80% | 8.45% | 8.45% | 8.45% | 31.24% |
| Fitness & Weight Loss | 34.34% | 45.45% | 39.39% | 53.54% | 30.30% | 31.31% | 59.60% | 43.43% | 10.10% | 23.23% | 22.22% | 35.72% |
| Technology & Gadgets | 27.87% | 47.54% | 45.90% | 39.34% | 24.59% | 31.15% | 60.66% | 24.59% | 9.84% | 9.84% | 16.39% | 30.70% |
| Cars | 35.19% | 40.74% | 46.30% | 46.30% | 25.93% | 22.22% | 61.11% | 22.22% | 12.96% | 11.11% | 14.81% | 30.81% |
| Celebrity News | 28.12% | 59.38% | 53.12% | 84.38% | 40.62% | 43.75% | 56.25% | 43.75% | 12.50% | 18.75% | 21.88% | 42.05% |
| Crafts | 31.11% | 47.78% | 50.00% | 44.44% | 21.11% | 28.89% | 63.33% | 26.67% | 8.89% | 17.78% | 13.33% | 32.12% |
| Painting | 24.51% | 44.12% | 46.08% | 53.92% | 27.45% | 34.31% | 57.84% | 33.33% | 13.00% | 18.63% | 14.71% | 33.45% |

你是一个精通中英双语、熟悉网络文化且极为严厉的翻译质量评估专家。你的任务是根据给定的上下文、原文和译文，按照以下严格的规则和格式进行评估。
1. 评分要求：
语义准确性核心要求：
  * 正确理解原文语义及情感：译文需准确反映原文的字面意思和隐含情感（如讽刺、兴奋、沮丧）。口语化表达里容易缺少标点符号，也容易出现因为断句理解错误而语义曲解。
  * 文化适应性称呼翻译：对"姐妹"、"老师"、"宝子"等称呼的翻译需考虑文化背景和社交礼仪，不能存在冒犯或歧义。
  * 符合目标文化语境：译文需符合英语文化语境和目标受众（社交媒体用户）的阅读习惯，避免中式英语。
  * 文化负载词处理：对承载特定文化、网络或语境含义的词汇（如"绝绝子"、"种草"），直译无法理解的，必须进行意译或文化替换，不可直译。
  * 单位换算准确性：涉及中式单位（如"亩"、"斤"、"里"）与国际单位的换算时，需准确并明确限定范围，避免歧义；对于中文习惯省略的单位，译文中需补充完整。
  * 专有名词准确性：对著名人名、地名、品牌名、作品名等特殊名词，要求翻译准确或使用公认译名。
翻译格式要求（一旦违反立即判0分）：
  * #占位x# 是不能翻译的，否则直接0分
  * 翻译前后<comment></comment> <title></title> <content></content>等标签无法对应，这意味着翻译结果无法正确被提取，违反这条直接0分

2. 评分标准：
你需要为整篇翻译给出一个 0到3分 的总体分数，定义如下：
* 0分：译文有严重错误，无法传递原文语义、内容丢失或曲解原文语义。无法让英文用户感受到原文的语境或者情绪。
* 1分：译文存在明显问题，但主要语义尚可被艰难理解。存在关键错误、文化误译，缺乏文化适应性，严重影响阅读体验。
* 2分：译文准确传达了原文的主要信息，语法基本正确，语境或情绪表达合理。
* 3分：译文精准、表达自然、符合英语文化语境，语法规范，传达原文的所有信息和情感，精准符合英语社交平台受众的阅读习惯。

3. 输出格式：
希望你既要输出思考的过程，也要进行一个总结，并给出最终的0到3的分数。回复的格式参考如下：

问题1位置：xxx
对问题1的评论：xxx
问题2位置：xxx
对问题2的评论：xxx
综上，对整句的翻译意见：xxx
最终分数：（只有数字）"

4. 评估示例：
* 示例1：
可参考的上下文：0.第一次看鸟片好紧张
1.#手养鹦鹉[话题]# #玄凤鹦鹉[话题]# #合法饲养[话题]# 尊滴好尴尬
2.昨天我家的狗子被不认识的大狼狗给咬噝了，那狗比我家狗足足大了2倍，看到的时候已经屁股对着屁股了，我老公说这时候不能去动他们，不然会出不来，受伤的，我家狗2条后腿都悬空着的[惊恐R]
原文：<comment3>必须引产，生不出来，太危险</comment3>
译文：<comment3>An abortion is necessary, it can't be born, it's too dangerous.</comment3>

输出：
问题1位置：An abortion is necessary
对问题1的评论："引产"是指人工诱发分娩以使胎儿存活，应译为"induced labor"或"an induction is necessary"。而"abortion"指终止妊娠，两者含义相反，此处为严重误译。
对整句的翻译意见：译文存在医学概念翻译的错误，导致语义误解。
最终分数：0

* 示例2：
可参考的上下文：无
原文：#OPPOFindN5 手感、续航、性能、防水、闪充、AI体验全面升级。
更多惊喜，2 月 20 日 19:00 发布会见！
译文：#OPPOFindN5 The feel, battery life, performance, water resistance, fast charging, and AI experience are all getting a major upgrade. And there's more to come! See you at the launch on February 20th at 7:00 PM!

对整句的翻译意见：无问题可指出
最终分数：3

* 示例3：
可参考的上下文：无
原文：<title>姐妹们快冲！这款面膜真的绝了！</title>
译文：<title>Sisters, go quickly! This facial mask is really absolute.</title>

输出：
问题1：go quickly! This facial mask is really absolute.
对问题1的评论："快冲！"在此处是强烈的购买建议，应译为"Run, don't walk!"或"You gotta get this!"。"绝了"表示极好，应译为"amazing"、"incredible"或"the best"。当前译法完全不符合英语种草文案的表达习惯。
问题2：Sisters
对问题2的评论：在非特定语境下，"姐妹们"作为一种亲切呼语，译为"Sisters"略显生硬。根据上下文，可简化为"Guys"或"Everyone"，或直接省略。
对整句的翻译意见：译文生硬直译，严重削弱了原文的号召力和情感色彩。
最终分数：1

5. 请你根据上述提示，对这个翻译内容以规定的格式进行评估，仅输出评估内容，不要输出其他无关内容。

{content}

输出：

*Figure 13.* The Prompt for Cultural Effectiveness Evaluation.

You are a highly rigorous translation quality evaluation expert with full proficiency in both Chinese and English, and with deep familiarity with internet culture. Your task is to evaluate a translation based on the given context, source text, and translated text, following the strict rules and format below.

1. Scoring Requirements
Core requirements for semantic accuracy:
• Correct understanding of meaning and emotion: The translation must accurately reflect both the literal meaning and the implicit emotions of the source text (e.g., sarcasm, excitement, frustration). Colloquial expressions often lack punctuation, and incorrect sentence segmentation may easily lead to semantic distortion.
• Cultural adaptation of forms of address: Translations of address terms such as "姐妹("girls")", "老师(teachers)", or "宝子(babes)" must consider cultural background and social conventions, and must not introduce offense or ambiguity.
• Conformity to the target cultural context: The translation should align with English cultural norms and the reading habits of the target audience (social media users), and must avoid Chinglish.
• Handling of culturally loaded expressions: Words or phrases that carry specific cultural, internet, or contextual meanings (e.g., "绝绝子(amazing)", "种草("recommend")") must be paraphrased or culturally adapted when literal translation would be unintelligible. Literal translation is not acceptable in such cases.
• Accuracy of unit conversion: When Chinese units (e.g., "亩(square meters)", "斤(1 jin= 0.5 kilograms)", "里(500 meters)") are involved, conversions to international units must be accurate and clearly specified to avoid ambiguity. Units commonly omitted in Chinese must be explicitly supplied in the English translation.
• Accuracy of proper nouns: Well-known names of people, places, brands, or works must be translated accurately or rendered using widely accepted standard translations.
Formatting requirements (violation results in an immediate score of 0):
• Placeholders such as "#占位x#(#Placeholder0#)" must not be translated. Translating them results in an immediate score of 0.
• If tags such as <comment></comment>, <title></title>, <content></content> do not correspond correctly before and after translation, the translated content cannot be correctly extracted; violating this rule results in an immediate score of 0.

2. Scoring Criteria
You must assign an overall score from 0 to 3 for the entire translation, defined as follows:
• 0 points: The translation contains severe errors, fails to convey the original meaning, distorts or omits key content, and does not allow English readers to perceive the original context or emotion.
• 1 point: The translation has obvious problems, but the main meaning can still be understood with difficulty. It contains critical errors, cultural mistranslations, or lacks cultural adaptation, seriously harming the reading experience.
• 2 points: The translation accurately conveys the main information of the source text, is mostly grammatically correct, and presents reasonable contextual or emotional expression.
• 3 points: The translation is precise, natural, culturally appropriate for English readers, grammatically correct, and fully conveys all information and emotions of the original text, matching the reading habits of English social media users.

3. Output Format
You are required to output both your reasoning process and a final summary, and then provide the final score from 0 to 3. The response format should follow the example below:

Issue 1 Location: xxx
Comment on Issue 1: xxx
Issue 2 Location: xxx
Comment on Issue 2: xxx
Overall translation feedback: xxx
Final Score: (number only)

4. Evaluation Examples

Example 1
Reference Context: 0. First time watching a bird video, so nervous 1.#Hand-raisedParrot[Topic]# #Cockatiel[Topic]# #LegalPetOwnership[Topic]# So embarrassing
2.Yesterday my dog was "that-ed" by a huge unfamiliar dog—twice her size. When we saw them, they were already butt to butt. My husband said we couldn't separate them or they'd get injured. My dog's two hind legs were completely off the ground [terrified]
Source Text: <comment3>必须引产，生不出来，太危险</comment3>
Translation: <comment3>An abortion is necessary, it can't be born, it's too dangerous.</comment3>

Output:
Issue 1 Location: An abortion is necessary
Comment on Issue 1: "引产" refers to inducing labor to deliver a viable fetus and should be translated as "induced labor" or "an induction is necessary." "Abortion" refers to termination of pregnancy, which has the opposite meaning. This is a severe mistranslation.
Overall translation feedback: The translation contains a critical medical terminology error that results in semantic distortion.
Final Score: 0

Example 2
Reference Context: None
Source Text: #OPPOFindN5 手感、续航、性能、防水、闪充、AI体验全面升级。更多惊喜，2 月 20 日 19:00 发布会见！
Translation: #OPPOFindN5 The feel, battery life, performance, water resistance, fast charging, and AI experience are all getting a major upgrade. And there's more to come! See you at the launch on February 20th at 7:00 PM!

Output:
Overall translation feedback: No issues identified
Final Score: 3

Example 3
Reference Context: None
Source Text: <title>姐妹们快冲！这款面膜真的绝了！</title>
Translation: <title>Sisters, go quickly! This facial mask is really absolute.</title>

Output:
Issue 1 Location: go quickly! This facial mask is really absolute.
Comment on Issue 1: "快冲！" expresses a strong purchase recommendation and should be translated as "Run, don't walk!" or "You gotta get this!" "绝了" means extremely good and should be rendered as "amazing," "incredible," or "the best." The current translation is unnatural and unsuitable for promotional English.
Issue 2 Location: Sisters
Comment on Issue 2: In this context, "姐妹们" is a casual form of address. Translating it as "Sisters" sounds awkward in English. Depending on context, "Guys," "Everyone," or omission would be more appropriate.
Overall translation feedback: The translation is overly literal and significantly weakens the original persuasive tone and emotional impact.
Final Score: 1

5. Task Instruction
Please evaluate the following translation content according to the above instructions and required format.
Only output the evaluation content. Do not output any other unrelated content.

{content}

Output:

*Figure 14.* The Translation Version of The Prompt for Cultural Effectiveness Evaluation.

**CULTURE-MT Case1**

Title：这一筐下去怕是要见太奶了🚑
Content：钻了一下午的小树林，累得够呛。结果拿回来一查，全是红伞伞白杆杆[笑哭R]。本来想尝鲜，现在变成绝命毒师了。这哪里是采蘑菇，简直是在排练躺板板[允悲R]。纯纯的大冤种，又是白干的一天！
#占位0# #占位1# #占位2# #占位3# #占位4# #占位5# #占位6#

**Qwen3-8B Translation of Case1**

Title: This basket might send me to meet Grandma 🚑
Content：Spent the whole afternoon poking around in the little woods, really exhausted. But when I checked back, it turned out to be all red umbrellas and white poles [sad crying]. I just wanted to try something new, but now it's like Breaking Bad. This isn't even foraging mushrooms—it's like rehearsing for a lying on the floor scene [tearful]. Pure innocent victim, another day wasted!
#占位0# #占位1# #占位2# #占位3# #占位4# #占位5# #占位6#                    Score：0

**Judger Response for the Translation of Case1 by Qwen3-8B**

**Issue 1 Location:** This basket might send me to meet Grandma
Comment on Issue 1: "见太奶" is an internet slang expression in Chinese, commonly used to humorously and hyperbolically refer to "meeting one's maker," "being doomed," or "kicking the bucket," often with a self-deprecating tone. The translation "meet Grandma" is a literal rendering (as "太奶" colloquially means "great-grandma"), but in English contexts, unless paired with phrases like "meet God" or "meet the Grim Reaper," the standalone phrase "meet Grandma" would confuse readers into thinking the speaker is literally visiting their grandmother. This results in a complete loss of the original's dark humor and exaggerated reference to death.
**Issue 2 Location:** red umbrellas and white poles
Comment on Issue 2: The original Chinese phrase "红伞伞白杆杆" vividly describes the appearance of poisonous mushrooms—typically species in the Amanita genus, such as the death cap (Amanita phalloides)—characterized by red caps and white stems. While "red umbrellas" superficially corresponds to "红伞伞," the cap of a mushroom is conventionally referred to as a "cap" in English mycological terminology. More critically, English readers encountering "red umbrellas" would likely picture actual rain umbrellas rather than mushrooms. An accurate translation should describe the mushroom's morphology, e.g., "red caps and white stems." The literal translation of "umbrellas" and "poles" is not only awkward but also botanically inaccurate.
Issue 3 Location: I just wanted to try something new, but now it's like Breaking Bad.
**Comment on Issue 3:** The original Chinese phrase "绝命毒师" is a pun: it refers both to the TV series Breaking Bad (in which the protagonist transforms from a chemistry teacher into a drug manufacturer) and functions as a self-mocking description of the blogger becoming an inadvertent "poison producer" after picking toxic mushrooms ("红伞伞"). Although the translation mentions Breaking Bad, it omits the crucial meaning of "绝命" (jué mìng), which conveys "deadly" or "fatal." More precisely, the original text humorously portrays the speaker as having become a "maker of deadly toxins," not merely someone resembling the TV character. While the translation is superficially fluent, it loses the core connotation of "poison." A more faithful rendering might be "now I've become a deadly poison dealer" or an adaptive translation that preserves the self-deprecating tone.
**Issue 4 Location:** rehearsing for a lying on the floor scene
Comment on Issue 4: The original phrase "排练躺板板" employs the internet slang "板板" , which refers to a coffin lid—a vivid, colloquial way to say "lying in a coffin." This is a culturally specific Chinese meme implying "preparing for one's own funeral." The translation "lying on the floor scene" entirely omits the reference to a coffin or death, reducing the meaning to a mundane act of lying on the floor. This constitutes a serious distortion of intent. A more accurate translation would be "rehearsing for my funeral" or "practicing for the coffin scene."
Issue 5 Location: Pure innocent victim
**Comment on Issue 5:** The original phrase "纯纯的大冤种" is a self-deprecating expression describing someone who has been thoroughly duped or suffered a great loss through foolishness—essentially a "total sucker" or "biggest fool." While "pure innocent victim" conveys victimhood, the word "innocent" in English typically implies being wronged by others' malice or crime, whereas the Chinese phrase emphasizes self-inflicted misfortune and comedic regret. More idiomatic equivalents would be "total sucker" or "such a fool."
**Overall Evaluation:**
The translation relies excessively on rigid literalism, resulting in substantial semantic loss and, in several cases, outright misinterpretation. The source text is rich in Chinese internet slang and culturally embedded humor centered on the blogger's near-fatal encounter with poisonous mushrooms. The translation fails to perform necessary cultural adaptation or functional equivalence, rendering the target audience unable to grasp the self-mocking humor and life-or-death exaggeration central to the original. While hashtags are handled correctly, the core semantics are severely compromised.
**Final Score: 0**

*Figure 15.* An example generated by Qwen-8B that is assigned a cultural effectiveness score of 0. The JUDGER provides a detailed and well-grounded evaluation explaining the judgment.

---

**CULTURE-MT Case1**

Title：这一筐下去怕是要见太奶了🚑
Content：钻了一下午的小树林，累得够呛。结果拿回来一查，全是红伞伞白杆杆[笑哭R]。本来想尝鲜，现在变成绝命毒师了。这哪里是采蘑菇，简直是在排练躺板板[允悲R]。纯纯的大冤种，又是白干的一天！
#占位0# #占位1# #占位2# #占位3# #占位4# #占位5# #占位6#

**Qwen3-8B Translation of Case1**

Title：This basket might send me to meet Grandma 🚑
Content：Spent the whole afternoon poking around in the little woods, really exhausted. But when I checked back, it turned out to be all red umbrellas and white poles [sad crying]. I just wanted to try something new, but now it's like Breaking Bad. This isn't even foraging mushrooms—it's like rehearsing for a lying on the floor scene [tearful]. Pure innocent victim, another day wasted!
#占位0# #占位1# #占位2# #占位3# #占位4# #占位5# #占位6#

Score：0

**Ours-32B Translation of Case1**

Title：After this basket, I'm pretty sure I'm going straight to meet grandma ambulance style🚑
Content：Spent the whole afternoon in that little woods, and I'm so beat. But when I got home and checked, yep, they were all those red-capped mushrooms with white stems[laugh crying R]. I was gonna try something new, but now I'm basically a death dealer. This wasn't mushroom foraging, it was basically a dress rehearsal for the board[so sad R]. Total sucker, another day for nothing!
#占位0# #占位1# #占位2# #占位3# #占位4# #占位5# #占位6#

Score：1

**Qwe3-32B Translation of Case1**

Title：This basketful might just get me to see Grandma in the afterlife ambulance 🚑
Content：Spent the whole afternoon foraging in the woods, totally exhausted. Only to find out later they're all red caps with white stems [tearful laughingR]. Initially wanted to try something new, now I'm turning into a death row chef. This isn't mushroom picking, it's basically rehearsing for lying on a stretcher [sad]. Pure big loser, another day wasted!
#占位0# #占位1# #占位2# #占位3# #占位4# #占位5# #占位6#

Score：2

**Ours-8B Translation of Case1**

Title：This basket is gonna get me 🚑
Content：Spent the whole afternoon in the woods and was totally wiped out. But when I got home and checked them, turns out they were all red caps and white stems [Cry with laughter]. I just wanted to try something new, but now I feel like I've become Walter White. This wasn't mushroom picking, it was basically me rehearsing for my final resting place [Sad]. Just a total sucker, and another day of work for nothing!
#占位0# #占位1# #占位2# #占位3# #占位4# #占位5# #占位6#

Score：3

*Figure 16.* Translation examples for the same case with scores ranging from 0 to 3, demonstrating varying degrees of cultural effectiveness.

