# OpenReview forum: "Beyond Literal Translation: Evaluating Cultural Effectiveness in Social Media UGC"
_ICML.cc/2026/Conference — ICML 2026 regular_

### Official Review · Reviewer_XfCp · 2026-02-27

**Soundness:** 3
**Presentation:** 3
**Significance:** 3
**Originality:** 3
**Overall Recommendation:** 5
**Confidence:** 4

**Summary:**

This paper proposes a UGC translation benchmark focused on cultural transmission and UGC-specific emotion resonance. It introduces a cultural effectiveness criterion and trains a related JUDGER model to jointly assess expression accuracy and cultural adaptability.

**Compliance With Llm Reviewing Policy:**

Affirmed.

**Final Justification:**

The author's rebuttal addresses my concerns raised in the initial official review, so I would recommend it as Accept.

**Key Questions For Authors:**

* Could you provide some intuitive examples to highlight the difference between your proposed benchmark and previous related online content translation benchmarks?
* Can you extend the benchmark to include more language pairs? Currently, it only covers Chinese-English, which feels somewhat limited.
* Could you provide more detailed information regarding the human annotations and explain how the reliability of the expert annotations is ensured?
* Can the JUDGER model be leveraged to assess other cultural-related content beyond translation tasks? I wonder if it could be generalized to other tasks to make a broader contribution.
* A more detailed explanation of the differences between your benchmark and previous benchmarks would be helpful in understanding its unique contribution.
* Could you elaborate on cultural effectiveness? Is it a simple true/false classification, or does it involve more detailed design? Is it assessed through human judgment or by setting prompts for large models? How do you ensure the feasibility of the cultural effectiveness metric? More experiments in this area would be beneficial.
* The references are limited, and there is a lack of related work, such as "Benchmarking Machine Translation with Cultural Awareness." Could you explain the differences between your benchmark and this one? Please include more citations to strengthen the background.

**Limitations:**

See questions and weaknesses

**Strengths And Weaknesses:**

### Strengths

* Expert annotations in constructing the benchmarks, with a machine-human collaboration workflow to ensure the reliability of the benchmark.
* A specifically designed JUDGER model to assess cultural appropriateness and translation performance for the benchmark suite.
* Well-written paper with clear experimental settings and results.

### Weaknesses

* The definition of cultural effectiveness is somewhat qualitative. I wonder if it can be automatically scaled.
* I am concerned that integrating the JUDGER model into training data augmentation may introduce training bias. Specifically, since the training data is generated from the JUDGER's grade (2-3), the data may be more biased toward the JUDGER's evaluation, potentially skewing results.

---

> ### Author Rebuttal · Authors · 2026-03-30
>
> We thank the reviewer for the positive feedback. We are encouraged that the reviewer recognizes the **reliability of our human–LLM collaborative annotation pipeline, the design of the JUDGER model for cultural evaluation, and the clarity of our experimental setup and results.**
>
> Below, we respond to the comments in detail.
>
> ---
>
> ### 1. Concern about “if the definition of cultural effectiveness can be automatically scaled,” and Question 4.
> Cultural effectiveness is defined through two broadly applicable dimensions—expressive accuracy and cultural adaptability—and can naturally extend to other generation tasks that require culturally appropriate outputs (e.g., social media generation or dialogue).
> We observe that JUDGER generalizes beyond CULTURE-MT (e.g., on Seed-X-Challenge and RedTrans), suggesting it captures a transferable evaluation signal.
>
> ---
>
> ### 2. Concern about "integrating the JUDGER model into training data augmentation may introduce training bias."
> Please see the response to  Reviewer 1cy3 **" For Question 3"**.
>
>
> ---
>
> ### 3. For Questions 1, 5, and 7: Difference from prior benchmarks
>
> We discuss this in Sec. 2.1 and note that prior work often focuses on specific phenomena or narrow settings, while the diversity of social media UGC remains underexplored. Below, we provide concrete comparisons.
>
> #### **(1) Seed-X-Challenge**
>
> Seed-X covers 18 domains (e.g., Internet, Office, Manufacturing, Finance) and multiple styles (e.g., colloquial, slang, formal, academic). In contrast, we focus on **14 high-impact UGC domains** aligned with real user attention, and cover most informal styles except *academic/literary*, which are less relevant to UGC.
>
> Key differences:
>
> * **Scale & density:** 32–102 samples per domain
> * **Compositional complexity:** multiple phenomena often co-exist in one Note
> * **Length:** **104.11 tokens** vs. **50.49 tokens** in Seed-X
>
> #### **(2) RedTrans-Bench**
>
> RedTrans targets social media but mainly uses simple source–reference pairs, e.g.:
>
> ```
> {
>       "ref": "Back to the 90s",
>       "src": "重返90年代",
>       "source": "note-content",
>       "src_lang": "zh",
>       "dst_lang": "en"
> }
> ```
>
> and does not provide explicit metrics for humor, memes, or emoji semantics.
>
> In contrast, we:
>
> * focus on **UGC notes** with richer interaction and style
> * define **cultural effectiveness**
> * provide a dedicated evaluator, i.e. **JUDGER**
>
> #### **(3) Other benchmarks**
>
> * **DITING:** long-form web novels
> * **TransBench:** structured domains (e-commerce, legal)
> * **Slang/Benchmarking Machine Translation with Cultural Awareness:** mainly lexical-level phenomena
>
> In contrast, we evaluate **holistic cultural effectiveness** (meaning, tone, and audience resonance; see the case in Fig. 15).
>
> We will add these analyses to the appendix to further clarify our contribution.
>
> ---
>
> ### 4. For Question 2: Can you extend the benchmark to include more language pairs?
> We thank the reviewer for this valuable suggestion.
> We agree that extending the benchmark to more language pairs is an important direction. In this work, we focus on Chinese–English as a representative and challenging setting, due to its significant cultural and linguistic differences, as well as the richness of UGC content.
> Importantly, our framework is not language-pair specific. The definition of cultural effectiveness (expressive accuracy + cultural adaptability), as well as the JUDGER-based evaluation pipeline, can be directly applied to other language pairs.
> We will extend our benchmark to additional language pairs (e.g., multilingual settings) in future work.
>
> ---
>
> ### 5. For Question 3: Could you provide more detailed information regarding the human annotations and explain how the reliability of the expert annotations is ensured?
> Please refer the reviewer to our response to Reviewer 1cy3 (“For Question 1 & Question 2”).
>
> ---
>
> ### 6. For Question 6: suggestion for more detial experiments for JUDGER.
> We thank the reviewer for this important question.
>
> (1) Metric definition.
> Cultural effectiveness is a 4-level scoring scheme (0–3) based on two dimensions: expressive accuracy and cultural adaptability (Table 6, Figure 14). We additionally report a binary grouping (ineffective vs. effective) for interpretability.
>
> (2) Reliability and implementation.
> Please refer to our response to Reviewer 1cy3 (“For Question 1 & Question 2”) for details on annotation reliability and JUDGER validation.
>
> (3) Additional evidence.
>
> As discussed in other responses, we provide:
> * ablation studies isolating the effect of the JUDGER objective (response about **“For Question 1 & Question 2”** for Reviewer 1cy3),
> * cross-benchmark evaluation (Seed-X, RedTrans),
> * and preliminary real-world evidence (user engagement improvements) from Response to Reviewer CtzW about Question 1.
>
> We will consolidate these analyses in the revision to make the evaluation more complete and easier to follow.
>
> ---
>
> We look forward to further discussion with you.

---

> > ### Author Rebuttal · Reviewer_XfCp · 2026-04-01
> >
> > Thank you for the authors’ detailed rebuttal.
> >
> > I also found a recent benchmark work (https://arxiv.org/abs/2509.21798) related to cultural aspects, which may be worth discussing in the related work section. More generally, it would be better to ensure that the related work is fully updated in the revision.

---

> > > ### Author Response · Authors · 2026-04-02
> > >
> > > I am glad to have clarified the issues you raised. And thank you for pointing out this relevant work. We will incorporate a discussion of CARB ( https://arxiv.org/abs/2509.21798 )  into the Related Work section of our revised manuscript.
> > >
> > > CARB evaluates **reward models' ability to judge culturally appropriate preferences** — specifically, whether a reward model can identify which response better aligns with a given culture's values, norms, and common-sense knowledge. Its data are drawn from structured, authoritative sources (Cultural Atlas, World Values Survey), and its evaluation targets reward models across 10 distinct cultures. Our CULTURE-MT, by contrast, evaluates **the effectiveness of cultural transmission in machine translation systems on real-world social media UGC** — specifically, whether a translation can convey culture-laden internet expressions such as buzzwords and platform-specific discourse styles in ways that resonate with target-language readers. Our data are sourced from authentic Chinese social media platforms and are characterized by informal, fragmented, and highly expressive linguistic styles that pose challenges distinct from those in structured cultural Q&A settings.
> > >
> > > Despite their different task orientations — CARB addressing cultural awareness in model alignment, and CULTURE-MT addressing cultural effectiveness in cross-lingual social media communication — the two works converge on a shared finding: **existing evaluation approaches tend to capture surface-level cultural signals rather than genuine cultural understanding**. CARB demonstrates that most reward models exhibit spurious correlations, relying on explicit cultural labels or linguistic form rather than substantive cultural content. Our work reveals a parallel limitation in translation evaluation: standard automatic metrics are similarly insensitive to cultural nuance, failing to distinguish literal translation from culturally effective translation.

---

### Official Review · Reviewer_wqLb · 2026-03-09

**Soundness:** 2
**Presentation:** 3
**Significance:** 2
**Originality:** 3
**Overall Recommendation:** 3
**Confidence:** 4

**Summary:**

The paper introduces a new Chinese-English machine translation benchmark, CULTURE-MT, specifically designed to evaluate translations of user-generated content (UGC). The benchmark covers translation cases with culture-loaded expressions and social media linguistic styles. Alongside the benchmark, the paper introduces a new evaluation metric based on LLM-as-a-judge to assess cultural effectiveness, which covers both translation accuracy and cultural adaptability. For this new metric, they trained an LLM called JUDGER, an automatic evaluator, to measure cultural effectiveness on a 0-3 scale. Using this benchmark, they run experiments with both closed-source and open-source LLMs. The authors also trained the  UGC Translation LLM baseline, which was specifically built by fine-tuning the Qwen model using a culturally effective translation corpus. Results showed that cultural effectiveness varies across domains and note types, that closed-source LLMs (GPT-5 and Gemini 3) still outperform open-source LLMs, and that the UGC translation LLM baseline can improve and perform on par with the best open-source LLMs.

**Compliance With Llm Reviewing Policy:**

Affirmed.

**Final Justification:**

As the discussion clearly shows, among all the weaknesses, I think only overfitting has been properly addressed. Regarding other concerns, in my opinion, they need to be addressed more thoroughly and require a significant update for the paper to reach a publishable state. Therefore, I retain my prior assessment because it has been reinforced throughout the reviewer-author discussion.

**Key Questions For Authors:**

1. Looking at the number of augmented data and the number of data that were used to train both UGC baselines and JUDGER, there are gap of around ~10K of data. Does this mean that the ~10K data are filtered by the cosine sim? Or are there any more explanations about this data?
2. Why did the authors choose to compare with only BLEU, chrF, and COMET? It is clear that these metrics are not intended to measure cultural adaptability, even though they are among the most commonly used MT metrics. However, given that the UGC is more likely to favor liberal translation than literal translation, I think the authors should have found better comparisons in more recent works on MT metrics.
3. How do the authors ensure that there is no single augmented data point that overlaps with the test data? Although it is understandable that it has a completely different generation pipeline, there could be some possibility in it, and better to be sure by explicitly checking them.
4. If eventually the new metric are more focused towards the share of ineffective vs effective samples, then why implement it as a 0-3 scale and not binary right from the start?
5. Section 3.3 states that it is based on two dimensions: expressive accuracy and cultural adaptability. Then, why not go fine-grained into each of these dimensions, rather than combine these two into one? I guess it will be better to interpret the results and give better feedback to improve the translation.

**Limitations:**

yes

**Strengths And Weaknesses:**

**Strengths**
1. The paper properly highlights relevant and underexplored challenges in machine translation (informal, culture and context heavy, etc.), which are tougher to evaluate.
2. The paper positioned itself well with other related works by highlighting its differences and unique approach (granularity of note types, broad coverage of verticals)
3. The benchmark dataset and its construction pipeline are novel contributions for machine translation, with involvement from both human experts and LLM, showing a great example of human and AI interaction that is able to produce such high-quality results.
4. A new cultural effectiveness metric that complements other machine translation metrics by measuring cultural resonance and adaptability of a translation, something that is obviously not covered in common MT metrics. The paper also showed that it is highly correlated with human judges, with reports on Cohen's Kappa value of 0.7205

**Major Weaknesses**
1. **Data augmentation and diversity issue**: In Appendix C, it is mentioned that they augmented the data by sampling with temperature=1 and top-p=1, and filtering using cosine similarity with a threshold of 0.75. I think this part is not quite robust, especially given the 0.75 semantic sim threshold, which is quite permissive and allows similar data to pass through. The fact that augmentation uses only Gemini 3 is also concerning, especially for JUDGER, which may introduce model-specific bias.
2. **Risk of Overfitting**: Following the previous issue, the UGC baselines and JUDGER are trained using the same sources of augmented data. This could raise an overfitting concern and undermine the authors' conclusion, since there is a circular dependency between the UGC baselines and JUDGER, which could explain why the UGC baselines perform significantly better than their base model.
3. **MT Metric Comparison**: The authors claim that the new metric is necessary because "standard automatic metrics are largely insensitive to cultural effectiveness". However, the comparison with other MT metrics is lacking. BLEU and chrF are both syntax- and lexically based metrics, whereas COMET measures general semantic similarity. None of them is intended to measure cultural adaptability. The Multidimensional Quality Metrics (MQM) [2] based metric, I think, is more suitable for comparison if the paper wants to show how significant the proposed metric is. There are some automated metrics based on MQM for comparison, such as GEMBA-MQM [3], Auto-MQM [4], or, even better, MQM-APE [5], which is complemented by post-editing. Also, there are LLM-based MT automated metrics, such as GEMBA [6] (a predecessor of GEMBA-MQM). If it's difficult to compare with these metrics, at least mention them since they may be related to the proposed metric, at least in terms of using LLM as evaluators for MT.

**Minor Weaknesses**
1. The data augmentation presentation is quite confusing and difficult to follow. For example, in the first column L289-L297, it mentions that it samples 3,000 instances from data in Section 3.2, but there are two different datasets constructed in 3.2 (1890 notes or the 1002 high-quality notes). Although I eventually understand that the 3,000 data are sampled from the 100K augmented data, it is unclear and confusing, especially since the augmentation details are in Appendix C and are only briefly mentioned in the second column, L205-L207.
2. There is a minor concern about how different the test data is from the augmented data, related to the diversity issues raised above. It would be better if the authors could provide more information about this.
3. There is an inconsistent show of how the cultural effectiveness metric should be interpreted. It was scored on a 0-3 scale, but the authors then heavily analyzed it as a binary "Ineffective" (0-1) vs "Effective" (2-3) by often using the "Ineff. (%)" rather than the avg. score.
4. Although the paper showed that it positively correlates with Cohen's Kappa of 0.7205, the paper does not address common biases associated with LLM-as-a-judge frameworks (such as model-specific bias mentioned before). Since the new metric relies entirely on JUDGER, it is better to provide a more detailed analysis of it. One reference for this is [1].

[1] Lee, C., Zeng, T., Jeong, J., Sohn, J. Y., & Lee, K. (2025). How to correctly report llm-as-a-judge evaluations. arXiv preprint arXiv:2511.21140.

[2] Lommel, A. R., Burchardt, A., & Uszkoreit, H. (2013). Multidimensional quality metrics: a flexible system for assessing translation quality. In Proceedings of Translating and the Computer 35.

[3] Kocmi, T., & Federmann, C. (2023, December). GEMBA-MQM: Detecting translation quality error spans with GPT-4. In Proceedings of the Eighth Conference on Machine Translation (pp. 768-775).

[4] Fernandes, P., Deutsch, D., Finkelstein, M., Riley, P., Martins, A. F., Neubig, G., ... & Firat, O. (2023, December). The devil is in the errors: Leveraging large language models for fine-grained machine translation evaluation. In Proceedings of the Eighth Conference on Machine Translation (pp. 1066-1083).

[5] Lu, Q., Ding, L., Zhang, K., Zhang, J., & Tao, D. (2025, January). MQM-APE: toward high-quality error annotation predictors with automatic post-editing in LLM translation evaluators. In Proceedings of the 31st International Conference on Computational Linguistics (pp. 5570-5587).

[6] Kocmi, T., & Federmann, C. (2023, June). Large language models are state-of-the-art evaluators of translation quality. In Proceedings of the 24th Annual Conference of the European Association for Machine Translation (pp. 193-203).

---

> ### Author Rebuttal · Authors · 2026-03-30
>
> We thank the reviewer for the thorough summary and for recognizing the significance and novelty of our work. Below, we address the questions and concerns in detail.
>
> ---
>
> ### 1. Concern about the diversity issue of training data
>
> **(1) For translation model: the cosine similarity threshold is used as a loose upper-bound filter to remove near-duplicate samples, rather than to enforce semantic closeness.** In practice, it allows substantial lexical and stylistic variation, which is critical for preserving UGC diversity. Moreover, the primary source of diversity stems from the human-authored metadata spanning 14 domains and 4 note types, rather than solely from LLM sampling.
>
> **(2) For JUDGER, we used both Gemini distilled data and human-annotated data.**  The strong agreement with human annotations (Cohen’s κ = 0.72) further suggests that JUDGER does not exhibit strong model-specific bias.
>
> ---
>
> ### 2. About the Risk of Overfitting for Baseline Models.
>
> Please see the response to  Reviewer 1cy3 **" For Question 3"**.
>
>
> ---
>
> ### 3. MT Metric Comparison
> Our goal is not to replace existing metrics (e.g., BLEU, COMET, MQM), but to address a gap not explicitly captured by them, namely, cultural resonance and UGC-specific adaptation.
>
> We thank the reviewer for highlighting the connection to MQM-based and LLM-based evaluation. **Our framework is related to MQM-style approaches (MQM and GEMBA-MQM) in using multi-dimensional criteria and error span detection (Table 6, Figure 13).** The key difference is that we explicitly model user-perceived cultural effectiveness in social media settings.
>
> We agree that MQM-based automatic metrics (e.g., AutoMQM, MQM-APE) are relevant. However, they are mainly designed for general-domain evaluation and often rely on complex prompting or post-editing pipelines, making them less suited to large-scale, noisy, and stylistically diverse UGC.
>
> In contrast, our approach defines a task-specific objective (cultural effectiveness), constructs aligned UGC data, and validates the evaluator with strong human agreement (κ = 0.72), supporting its reliability in this setting.
>
> **We will include a reference and clearer comparison with MQM-style methods in the revision.**
>
> ---
>
> ### 4. Clarification on data augmentation and train–test separation
> Our pipeline involves three distinct datasets:
>
> (1) **Meta data (1,890)**: manually constructed;
>
> (2) **Benchmark data (1,002)**: filtered and enriched from meta data for evaluation;
>
> (3) **Augmented data (~100K)**: LLM-generated from meta data for training.
>
> The 3,000 expert-labeled samples for JUDGER are drawn from the augmented data, not from the meta or benchmark sets.
>
> **Train–test separation.**
>
> The benchmark and augmented data are constructed through different processes and remain strictly separated. The benchmark set is carefully curated with human verification, while the augmented data is generated from metadata with only weak guidance (topics and limited style cases). It does not reuse or paraphrase benchmark samples.
>
> Accordingly, augmented data is used for training (JUDGER and translation models), while the benchmark set is used exclusively for evaluation, ensuring no data leakage.
>
> We will clarify this more explicitly in the revision.
>
> ---
>
> ### 5. Response to Metric Interpretation Consistency
> Both the 0–3 scores and the binary Ineff. (%) metrics are consistently reported (e.g., Table 4 includes average scores and full distributions). We agree, however, that our analysis emphasizes the binary view, which may appear inconsistent. The 0–3 score is the primary metric, providing a fine-grained assessment, while the binary split (0–1 vs. 2–3) serves as a task-oriented view of whether a translation is usable for cross-cultural communication.
>
> ---
>
> ### 6. Response to JUDGER Bias and Reliability
> Please reference the response about **“For Question 1 & Question 2”** for Reviewer 1cy3。
>
> ---
>
> ### 7. Response to Data Usage and Count Discrepancy
> The augmented dataset (~100K) is split into two **disjoint subsets**:
> * **JUDGER training:** ~40K synthetic + 3K expert data, from which a 30K score-balanced set is constructed.
> * **Translation training:** ~52K augmented notes, refined (≤3 iterations) into ~50K high-quality instances.
>
> These subsets are **non-overlapping**, ensuring no data leakage between evaluator and model training. We will clarify this data split and usage in the revision.
>
> ---
>
> ### 8.Response to Granularity of Evaluation Dimensions
> We agree that separately analyzing expressive accuracy and cultural adaptability can provide more detailed insights.  However, in many UGC cases, errors in these two aspects are closely coupled, and separating them does not always lead to a clearer judgment of usability. Therefore, we report a unified score as the primary metric, while treating the two dimensions as underlying criteria for evaluation.
>
> ---
>
> We look forward to further discussion with you.

---

> > ### Author Rebuttal · Reviewer_wqLb · 2026-04-02
> >
> > We thank the authors for their detailed responses. However, several of our core concerns remain unaddressed or insufficiently resolved, and we elaborate below.
> >
> > **Data Diversity:**
> >
> > I find the argument that the diversity came from the human-authored metadata and further amplified by multi-sampling is insufficient to address my concern. First, 14 domains x 4 notes structure only provides topic coverage at a coarse level, but does not guarantee semantic diversity in fine-grained level. For example, there are 2 closely related domains in the 14 included, *Sports* and *Fitness & Weight Loss*, and there is no clear mechanism to guarantee that LLM do not generate same topics and notes. There is also no guarantee that given the same note it will not generate the same topic. The argument that cosine sim are used only for neardup filtering also emphasize this issue, allowing semantically similar notes with only difference in lexical choices and styles. What I am expecting from this is a kind of analysis of diversity from the whole 100K augmented notes, for example, topic model coverage, unique cultural expression counts, topic/note cluster, or anything that can prove the claim that it is diverse enough to be used in both training the baseline and training JUDGER.
> >
> > **Overfitting and Circular Evaluation**
> >
> > The provided ablation does not answer my concern. My concern is about evaluation bias due to shared data source, since both JUDGER and translation baseline derived from the same Gemini 3 augmented data. This creates a risk that JUDGER implicitly learned to prefer outputs similar to what Gemini 3 generated, which the translation baseline also more likely to produce because its trained on it too. The reports on high Cohen's k between JUDGER and human judges done on a general sample, not specifically the translation baseline, so its partially address the concern but not completely.
> >
> > **MT Metric, MQM**
> >
> > I respecfully disagree with the authors' response that MQM-based metrics are *"mainly designed for general-domain evaluation"* and involve *"complex prompting or post-editing pipelines, making them less suited to large-scale, noisy, and stylistically diverse UGC."*. I stated my reasons below:
> > 1. The original MQM Framework [2] is explicitly designed as a flexible, extensible quality metric that includes sub-dimensions such as style, locale convention, terminology, and fluency. These dimensions map directly onto what cultural effectiveness aim to measure. The authors did not provide any reliable evidence to their claim on why MQM is not suited for UGC, therefore I cannot accept the claim. If one of the argument is "complex prompting", then I can also argue the JUDGER prompt is quite complex looking at the provided prompt in the appendix.
> > 2. More fundamentally, the concern is not whether MQM is easier or harder to apply. My concern is whether the paper can prove that cultural effectiveness measures something **orthogonal** to what existing MT metric, specifically MQM, already captures. That is why I have a major concern comparing syntax-based metric with cultural-effectiveness, because showing divergence from these metrics does not establish necessity for cultural effectiveness relative to the availability of MQM-style approaches.

---

> > > ### Author Response · Authors · 2026-04-07
> > >
> > > We sincerely thank the reviewer for the careful follow-up and for highlighting these important concerns. Your comments helped us identify several points that were not explained clearly enough in the current draft. We respond to the three issues below.
> > >
> > > #### **1. On data diversity**
> > > We agree with the reviewer that coarse-grained coverage by 14 domains × 4 note types alone is not sufficient to establish fine-grained semantic diversity. To better address this concern, we further conducted topic clustering within each taxonomy, rather than clustering the full corpus only at the global level. **This analysis shows that diversity is not merely across domains, but also within each domain.**
> > >
> > > The results are in https://anonymous.4open.science/r/ICML_Rebuttal-CULTURE-MT/EN_Cluster_for_each_taxonomy.md
> > >
> > > For example, within Sports, the clusters further separate into running and marathon logs, badminton partner-seeking, swimming practice, tennis, football league discussion, table-tennis fandom and match-related content, basketball/NBA discussion, boxing/combat sports, and billiards/golf. Within Outdoor, the data further split into skiing, camping, hiking, mountaineering, cycling, diving, surfing, rock climbing, fishing, beachcombing, drifting, and paddleboarding. Within Home & Decoration, the clusters distinguish renovation/design, full-house customization, kitchen/bathroom organization, furniture and bedding, curtains/windows, home appliances, fragrance/flowers/tea ware, and maintenance-related topics such as waterproofing and formaldehyde removal. Similarly, within Pets, the clusters include not only cats and dogs, but also adoption/rescue, pet services, pet food and health products, and various small or exotic pets such as birds, rabbits, hamsters, turtles, and ornamental fish.
> > >
> > > **These results suggest that our augmented corpus not only covers broad topical categories, but also contains rich fine-grained semantic subgroups within each category.** In other words, the diversity is not simply reflected by having many coarse domains, but by the fact that each domain further branches into distinct subtopics, user intents, and expression scenarios.
> > >
> > > We will incorporate this analysis into the revised version to better substantiate the diversity of the augmented data.
> > >
> > > #### **2. On overfitting and circular evaluation**
> > >
> > > We thank the reviewer for raising this concern, as it also revealed an important omission in our description. The **43K raw instances used for JUDGER training are not translated in the same way as the data used for translation-model training**. Instead, we split the data into multiple groups and translated them using **multiple models listed in Table 2**, rather than relying on Gemini alone. In this pipeline, Gemini is mainly used in the **JUDGER scores annotation-related stage**, rather than as the sole source of translations.
> > >
> > > Moreover, after we balance the JUDGER training data across the 0–3 score labels, many high-scoring examples, especially 3-score outputs from stronger models, are further filtered out. As a result, the final JUDGER training set does not simply preserve or reinforce a preference for Gemini-style outputs. We agree that this point is very important, and we will make it explicit in the revised paper to avoid the impression of circular evaluation.
> > >
> > > #### **3. On MQM and related MT metrics**
> > >
> > > We would first like to clarify our intended meaning more precisely. We did **not** mean that **MQM itself** is inherently too complex or unsuitable for UGC. Rather, our point is that **MQM-derived evaluation methods often require substantial task-specific design choices** before they can be effectively adapted to a new setting such as cultural effectiveness evaluation. In this sense, we agree that our own method is also complex, and indeed it is explicitly **inspired by MQM-style structured assessment**.
> > >
> > > This connection can already be seen from the judging dimensions and criteria shown in **Figure 13 and Figure 14**, and we will add clearer discussion and citations to MQM and its derivatives in the revised version.
> > >
> > > We also note that our additional experiments on **Gemini** and **Qwen3-32B** with an **MQM-style prompt** already serve as a direct comparison to MQM-based evaluation. Specifically, Gemini + MQM-style prompting achieves 89% accuracy, while Qwen3-32B + the same MQM-style prompting achieves 78% accuracy. In comparison, our trained JUDGER reaches 86% accuracy. **We believe these results show that MQM-style prompting is indeed a strong foundation, while JUDGER’s gains do not come merely from prompt wording, but from task-specific training for cultural effectiveness evaluation.**
> > >
> > > We will revise the paper accordingly to make these points clearer and more precise. We again sincerely thank you for the thoughtful and constructive feedback. **Should you find that our revisions and clarifications have resolved your remaining concerns, we would greatly appreciate your reconsideration of the score.**

---

### Official Review · Reviewer_CtzW · 2026-03-13

**Soundness:** 3
**Presentation:** 3
**Significance:** 3
**Originality:** 4
**Overall Recommendation:** 3
**Confidence:** 3

**Summary:**

This paper addresses a meaningful and practical problem in social media translation: the handling of culture-laden expressions. The paper introduces CULTURE-MT, a new benchmark designed to evaluate the cultural effectiveness of translations for user-generated content (UGC). To support this, authors propose a novel evaluation criterion, cultural effectiveness, which emphasizes that a translation should not only be semantically accurate but also evoke a comparable emotional and contextual response in target-language users. Based on this criterion, they develop JUDGER, an automatic evaluator trained using a combination of expert annotations and LLM-generated data, and demonstrate its strong agreement with human judgments.

**Compliance With Llm Reviewing Policy:**

Affirmed.

**Key Questions For Authors:**

- While CULTURE-MT is an innovative and inspiring piece of work, its validity may be confined to the "cultural world" defined by its expert-LLM construction pipeline. For example, the understanding of Gen Z internet slang can vary significantly even among experts of different ages. How closely does your constructed standard align with the diverse, dynamic, and authentic culture of global social media?

- How can you ensure that JUDGER is not merely learning to replicate the standards of the LLM-human feedback loop on which it was trained? While its performance on the CULTURE-MT test set is validated, can it reliably assess entirely new cultural phenomena or emerging internet memes not present in its training data?

- Is the cultural effectiveness metric inherently dependent on contextual information? If so, how should context be incorporated into the evaluation framework?

**Limitations:**

Yes

**Strengths And Weaknesses:**

# Strengths:
- Soundness: The paper presents a systematically designed experimental framework. The methodology is well-suited to the research question and effectively demonstrates the limitations of traditional metrics in capturing cultural nuance. The results  show that cultural effectiveness can be improved through targeted training.
- Presentation: The paper is clearly written and easy to follow.
- Significance: The research addresses a highly relevant and practical problem for real-world social media translation scenarios.
- Originality: The work introduces a novel metric and benchmark specifically designed to measure the cultural resonance of translated social media content.
# Weaknesses:
-  Quantifying "cultural effectiveness" is an inherently difficult problem. The paper's approach of defining this multifaceted concept through the subjective judgments of a small group of experts, combined with LLM-synthesized data, seems unrealistic for capturing the diverse, dynamic nature of real-world cultural resonance.
- The proposed metric appears to be context-dependent, yet this is not addressed in the experimental design. For instance, the "culturally effective" translation provided in the example on lines 236-240 is confusing when taken out of context.
- The lack of a plan to release code and data raises concerns about reproducibility.

---

> ### Author Rebuttal · Authors · 2026-03-30
>
> We thank the reviewer for recognizing the novelty and practical relevance of our work, particularly in addressing cultural resonance in UGC translation beyond lexical-level phenomena.
>
> A central concern raised is whether our benchmark and evaluation framework reflects a realistic and generalizable notion of cultural effectiveness, or is instead limited to the “cultural world” defined by our annotation pipeline.
>
> > **Q1**: While CULTURE-MT is an innovative and inspiring piece of work, its validity may be confined to the "cultural world" defined by its expert-LLM construction pipeline. For example, the understanding of Gen Z internet slang can vary significantly even among experts of different ages. How closely does your constructed standard align with the diverse, dynamic, and authentic culture of global social media?
>
> We do not aim to define a fixed or universal notion of culture. Instead, as described in Sec. 3.3, cultural effectiveness evaluates whether a translation enables correct interpretation and comparable user experience, rather than enforcing specific expressions or styles.
>
> CULTURE-MT spans 14 domains and four note types (General, Express, Symbol, Hybrid), covering diverse linguistic styles and cultural phenomena, which helps mitigate bias from any single annotator perspective.
>
> Beyond offline evaluation, **we observe that models trained with JUDGER-guided data improve dwell time and click-through rates in a real platform setting**, suggesting that the metric correlates with actual user engagement rather than a closed annotation standard.
>
> > **Q2**: How can you ensure that JUDGER is not merely learning to replicate the standards of the LLM-human feedback loop on which it was trained? While its performance on the CULTURE-MT test set is validated, can it reliably assess entirely new cultural phenomena or emerging internet memes not present in its training data?
>
> JUDGER is trained on a combination of 40K LLM-annotated and 3K expert-labeled data, with score-balanced sampling and a structured rubric (Sec. 3.3), rather than a single feedback.
>
> On held-out evaluation data, JUDGER achieves 86.03% accuracy and κ = 0.72, showing strong agreement with human annotations, which indicates it captures general evaluation criteria rather than memorizing labels.
>
> As shown in Table 5, **JUDGER provides consistent discrimination across multiple benchmarks (CULTURE-MT, Seed-X-challenge, RedTrans), suggesting it is not tied to a single dataset.**
>
> We agree that rapidly emerging cultural phenomena remain challenging. Our goal is to provide a scalable approximation of human judgment, rather than a fully universal evaluator.
>
> > **Q3**: Is the cultural effectiveness metric inherently dependent on contextual information? If so, how should context be incorporated into the evaluation framework?
>
> We focus on UGC Notes where key context is embedded in the text (e.g., linguistic style, cultural symbols, and discourse patterns), making text-based evaluation feasible.
>
> As clarified in our response on data filtering, we remove samples requiring external multimodal context, ensuring that each instance can be meaningfully evaluated.
>
> We agree that a richer context (e.g., multimodal signals or user-level information) is important, and extending the framework in this direction is a key future direction.
>
> > **Weakness3**: The lack of a plan to release code and data raises concerns about reproducibility.
>
> We agree that reproducibility is important and will release:
> * the CULTURE-MT dataset,
> * and the JUDGER model (or evaluation interface),
> * and the Baseline Translation Models trained by us.
>
> ---
>
> **Conclusion:**
>
> Overall, we position cultural effectiveness not as a fixed or universal concept, but as a **practical and scalable evaluation objective**.
>
> ---
>
> Through diverse data construction and preliminary real-world evidence mentioned in Q1, we believe our framework provides a meaningful step toward culturally aware UGC translation.
>
> We look forward to further discussion with you.

---

> > ### Author Rebuttal · Reviewer_CtzW · 2026-04-03
> >
> > Thank you for the detailed rebuttal. It clarifies several important points and improves my understanding of the paper.
> >
> > I appreciate the clarification that cultural effectiveness is intended as a practical and scoped objective rather than a universal definition. The additional details on dataset diversity, training setup, and cross-benchmark evaluation also help address concerns about robustness. The plan to release data and models resolves my reproducibility concern.
> >
> > However, I still have some reservations. In particular, it remains unclear how well JUDGER can generalize to truly new or rapidly evolving cultural phenomena (e.g., emerging slang or memes). In addition, while you clarify that context is mostly embedded in the text, the dependency on context and its limitations could be more explicitly discussed.
> >
> > Overall, my concerns are partially addressed, but some questions about generalization and scope remain.

---

> > > ### Author Response · Authors · 2026-04-03
> > >
> > > We sincerely thank the reviewer for the insightful comments and are glad to address your further questions. We provide detailed clarifications regarding your two main concerns below:
> > >
> > > ---
> > >
> > > ### **Q1: Can JUDGER adapt to emerging internet phenomena?**
> > >
> > > To validate this, we extracted a trending post on today's social media platforms related to **"ICML2026 Rebuttal"**. We translated this post using Qwen3-32B, GLM-5, and our fine-tuned Ours-32B, and subsequently evaluated the translation results using JUDGER.
> > >
> > > The detailed evaluation results are available at: https://anonymous.4open.science/r/ICML_Rebuttal-CULTURE-MT/Case_Study_for_JUDGER.md
> > >
> > > **Why does this support adaptability to new phenomena?**
> > > Notably, the expressions in this example are highly time-sensitive and community-specific, centered on the recently discussed topic of ICML rebuttal score changes. Although such expressions may not appear in standard translation benchmarks, JUDGER still provides fine-grained feedback on score semantics, internet-specific wording, and stylistic naturalness, and assigns a higher score to the translation that better preserves the original online tone. This suggests that JUDGER’s evaluation ability is not limited to pre-defined cultural symbols, but extends to newly emerging discourse patterns in real social-media environments.
> > >
> > > ---
> > >
> > > ### **Q2: In-depth Discussion on "Context"**
> > >
> > > We greatly appreciate your insightful observation. As noted in our previous response, the note data selected for this study was chosen specifically because it does not rely on context. However, we recognize that context plays a critical role in other formats, such as video notes, brief notes, and comments. **Given that our primary focus in this paper is cultural effectiveness, we prioritized retaining posts that embody richer internet cultural phenomena.**
> > >
> > > **Regarding the incorporation of "context," this was already taken into account during the design of the JUDGER prompt. As illustrated in Figures 13 and 14**, our examples incorporate "context references," including samples of short notes and comments. This aligns with our immediate next step: constructing a social media translation evaluation system that necessitates context references.
> > >
> > > We appreciate your continued attention to this issue, which has helped us recognize its value. We will incorporate a discussion on "context" in the revised version and expedite the progress of this follow-up work. **Should you find that our revisions and clarifications have resolved your remaining concerns, we would greatly appreciate your reconsideration of the score.**

---

### Official Review · Reviewer_1cy3 · 2026-03-16

**Soundness:** 2
**Presentation:** 1
**Significance:** 2
**Originality:** 3
**Overall Recommendation:** 3
**Confidence:** 3

**Summary:**

This paper introduces a Chinese to English translation benchmark spanning with a focus on content styles (like social media slang) which can help ascertain the cultural effectiveness of the translation in addition to the translation accuracy. They also contribute a trained a judgment model for automatic evaluation. The paper focuses on context that existing MT benchmarks largely miss: not just slang + culture-loaded symbols, but also the rhetorical, expressive, and interactional styles that make social media text feel socially native. Their experiments on this dataset help the authors conclude that there is some correlation between the size of the model and its cultural competence and that cultural modeling can enhance cultural effectiveness robustly.

**Compliance With Llm Reviewing Policy:**

Affirmed.

**Final Justification:**

The detailed rebuttal by the authors clarified several implementation details but two of main concerns still seem unresolved.

A minor point is that the authors's rebuttal acknowledgment stated that Qwen+JUDGER+Gemini statistics; For Qwen and Judger, I believe the performance being pointed out is in Table 3 (would request authors to highlight this), but in the submitted materials I still cannot locate the rebuttal’s Gemini + MQM-style result, which makes that comparison difficult to assess. Even if I am comfortable trusting that number, overall, I do not think it's wise to claim implicit evaluation on this baseline especially when its pitfalls have been highlighted in the main draft.

I think a more pertinent point for me is the 2nd concern: the benchmark construction still applies an LLM-driven cross-lingual value filter **before** enrichment, retaining only content judged valuable for non-Chinese audiences.  Because cultural enrichment happens after this filtering step, it does not fully address my concern that culturally local, covert, or hard-to-transfer content may already have been excluded upstream. Because benchmarks are static artifacts that impact generally the entire community, I do think that such design decisions are unfortunately more critically evaluated and in this case, I do think that this is a strong design limitation.

Overall, I appreciate the paper’s merits and the authors’ clarifications, but these remaining issues are substantial enough that I am maintaining my original score. I am reducing my confidence score though because I acknowledge that authors have taken **a lot** of effort (including multiple rounds of potentially costly human evaluation) to build this benchmark and if other reviewers + ACs believe that this design decision is not a critical pitfall, I will support the utility of the work.

**Key Questions For Authors:**

1. Can you report inter-annotator agreement for the 3,000 expert-labeled set, and also agreement/correlation between the 40,000 Gemini annotations and that expert gold set?

2. Can you report some correlation between the 40,000 Gemini-annotated samples and the expert-labeled set before using them for JUDGER training? Right now it is hard to tell how much noise is being imported into the evaluator through the synthetic labels.

3. How much of the gain from the 8B/32B models that you've trained is coming from the cultural-effectiveness objective specifically, versus simply training on ~50K more in-domain rewritten examples? An ablation removing the JUDGER-guided rewrite loop would help make this clearer.

**Limitations:**

No. The authors have clearly taken a lot of efforts to do 2 rounds of human supervision and while costs are discussed in detail in the appendix - details that can clearly substantiate the quality of the human annotation are not discussed clearly. This also confounds the limitations of the work as the benchmarks' validation is strongly dependent on the quality of the human annotation.

**Strengths And Weaknesses:**

Strengths:

1. Genuine gap being addressed with practical relevance: The fact that the benchmark is not just focusing on slang but works to include culture-loaded symbols plus expressive / rhetorical / interactional styles, which does feel meaningfully underexplored in prior work.


Weaknesses:

1. The 'BLEU/Comet/chrf metrics are not culturally sensitive result' is not surprising enough on its own: The authors already acknowledges that these metrics were not designed for this construct and also cite stronger, more specialized alternatives in related work. So, the claim about the insufficiency of these metrics and the consequently utility demonstration of the author' own proposed JUDGER, seems ineffective in comparison because these other metrics are clearly bad baselines.

2. The benchmark construction has a pretty strong normative bias already ?: using an LLM to retain only content that is 'valuable for non-Chinese audiences' seems like a major source of bias/inconsistency, especially for a paper whose whole claim is about preserving culturally loaded meaning. There is a real chance that overly local/culturally overloaded content gets filtered out before evaluation even begins. I could not find an analysis of what this filter removes or how much it changes the data distribution.

3. The human–LLM collaboration details are too underspecified for something this subjective: the paper says five open-source LLMs generate translations and human translators then 'synthesize' them into a final refined version, but it is not clear what 'synthesize' means in practice. Are annotators selecting one candidate, editing one, combining multiple, or effectively doing a fresh human rewrite? Is the final reference a mashup or some kind of adjudicated (Like a majority vote) choice? These details matter a lot here and do not seem to be spelled out.

Minor comment: Figure 8 is genuinely hard to read. With that many grouped bars, it is difficult to make out which methods are consistently good; hatching / faceting / stronger contrast would help a lot.

---

> ### Author Rebuttal · Authors · 2026-03-30
>
> Thank you for the detailed and valuable feedback and for **recognizing the core motivation of our work**. As highlighted, our benchmark goes beyond prior work focused on slang by jointly capturing culture-loaded symbols and expressive, interaction-driven styles in UGC. **It is an under-explored yet practically relevant direction.**
>
> Below, we provide detailed responses to the reviewer’s concerns.
>
> ---
>
> ### 1. Concern regarding the validity of the JUDGER metric.
> We agree that BLEU, chrF, and COMET are not designed to capture cultural effectiveness; our goal is not to question their general validity, but to highlight their misalignment with our objective, which focuses on cultural resonance and pragmatic adaptation in UGC rather than lexical or semantic similarity. We also agree that MQM-based and LLM-based metrics are relevant comparisons. **We provide a detailed discussion in our response to Reviewer wqLb (“MT Metric Comparison”) and will include this clarification in the revision.**
>
> ---
>
> ### 2. Clarifying concerns about potential bias from the “cross-lingual value” filtering
>
> **(1) Filtering objective.**
> The “cross-lingual value” criterion (Lines 245–248, Figure 9) is designed to focus on the content that is relevant, informative, **or** appealing to non-Chinese audiences. With “translation feasibility” criteria together, the goal is to ensure that each instance can be meaningfully evaluated for cultural effectiveness, rather than to favor specific content types.
>
> **(2) No evidence of cultural bias.**
> Retention rates are: General 43.1%, Express 49.6%, Symbol 50.2%, Hybrid 69.5%. The filtering actually removes more General samples, which typically lack strong cultural or stylistic signals.
>
> **(3) Removed samples are low-information or non-evaluable.**
> Among removed data, 56.4% require multimodal context, and 43.6% are short, low-content posts (e.g., “晨跑+网球🎾”). Consistently, average length increases from 71.17 to 104.11 tokens (+46.3%) after filtering, indicating removal of sparse content.
>
> **(4) Cultural coverage is preserved.**
> The final benchmark still contains 57.2% Symbol/Hybrid samples (573/1002) across 14 domains (Table 1), demonstrating strong coverage of culturally rich and stylistically complex content.
>
> We will include this analysis in the revision.
>
> ---
>
> ### 3. Clarifying the human–LLM collaborative annotation protocol
>
> “Synthesis” refers to structured adjudication rather than free-form rewriting. Annotators select the best LLM candidate as a base and refine it for cultural adaptation and fluency, optionally incorporating phrases from others. Rewriting from scratch is avoided unless all candidates fail.
>
> This design balances diversity (multiple LLM candidates) with consistency and quality control (human refinement).
>
> We will clarify this protocol and add examples in the appendix for transparency and reproducibility.
>
> ---
>
> ### 4. For Question 1 & Question 2: annotation reliability and synthetic label noise for JUDGER
>
> We thank the reviewer for raising this important concern.
>
> Two experts independently labeled all evaluation samples (84 × 11), achieving 72% agreement on 4-class labels and 93% on binary labels. All disagreements were resolved via third-party adjudication, ensuring high consistency. (The 3,000 training annotations are partitioned and thus do not report agreement.)
>
> Gemini annotations show strong agreement with expert labels, with 88.00% accuracy and Cohen’s κ = 0.76, indicating reliable large-scale supervision rather than significant noise.
>
> We further reduce noise by combining synthetic and expert data (40k + 3k), applying score-balanced sampling, and training under a rubric prompt (Sec. 3.3). The final JUDGER achieves strong agreement with human annotations (Table 3), validating its reliability.
>
>
> ---
>
> ### 5. For Question 3: An ablation removing the JUDGER-guided rewrite loop
> We thank the reviewer for this insightful question. We provide an ablation study to disentangle the effect of the JUDGER-guided objective from in-domain data alone：
> | Models  | Ineff. ↓ | Eff. ↑ | 0     | 1      | 2      | 3      | AVG. |
> |--|--|--|--|--|--|--|--|
> | Deepseek V3.2  | 16.57%  | 83.43%  | 3.09%  | 13.47%  | 58.08%  | 25.35%  | 2.06  |
> | Ours-32B   | 13.97%   | 86.03% | 1.40% | 12.57% | 58.18% | 27.84% | 2.12 |
> | Ours-8B   | 14.47%   | 85.53% | 0.50% | 13.97% | 57.29% | 28.24% | 2.13 |
> | Ours-32B w/o JUDGER loop  | 16.87%   | 83.13% | 2.40% | 14.47% | 56.49% | 26.65% | 2.07 |
> | Ours-8B w/o JUDGER loop   | 16.17%   | 83.83% | 1.20% | 14.97% | 56.69% | 27.15% | 2.10 |
>
> Training on ~50K in-domain data alone already yields strong performance, with w/o JUDGER models comparable to strong baselines (e.g., DeepSeek V3).
>
> Adding the JUDGER-guided rewrite loop further improves performance, reducing Ineff. by 2.9% (32B) and 1.7% (8B), while increasing the proportion of score-3 outputs.
>
> Overall, in-domain data provides a strong foundation, and the JUDGER-guided objective brings additional, targeted gains.

---

> > ### Author Rebuttal · Reviewer_1cy3 · 2026-04-03
> >
> > Thank you so much for running all the additional experiments and substantiating the claims with additional details; Most of my concerns are resolved provided these details (synthesis paradigm and agreement statistics specifically) are included in the final draft.
> >
> > I am still not convinced with the substantiation for the following:
> > 1. I still think the metric-comparison case would need a stronger treatment in the paper itself, since the current framing against BLEU/chrF/COMET does not by itself fully validate the utility of JUDGER. I appreciate that the authors enumerate the limitation of the MQM based methods being too-general domain but just their general omission makes the analysis still weak, in my opinion as these other baselines are too obviously deficient.
> >
> > 2. Cross-lingual filtering is content-type agnostic: According to your current response, the cross-lingual filtering is done over any content that is deemed relevant, informative, or appealing all of which are again pretty subjective. Combining that with the content having the potential for "cultural effectiveness" (which is also subjective) just seems to exacerbate this. The General/Express/Symbol/Hybrid taxonomy produced as a part of this also seems arbitrary so that even though the retention rates are informative, it doesn't clarify if culturally local or difficult-to-transfer content is not being systematically filtered out before evaluation.
> >
> > Nit Pick Note (Just realized while re-reviewing: Ln 187 - the spelling of Express is incorrect. Insignificant typo to point out for revision).
> >
> > Accordingly, I will maintain my score.

---

> > > ### Author Response · Authors · 2026-04-07
> > >
> > > We sincerely thank the reviewer for the thoughtful follow-up comments and for recognizing the additional experiments and clarifications we provided. We are encouraged that most of the concerns have been resolved. Below, we briefly address the two remaining points.
> > >
> > > ### **1. On the comparison between JUDGER and MQM-based methods**
> > >
> > > Thank you for this important suggestion. In fact, our evaluation prompt design is itself **based on MQM-style structured assessment**, which can also be seen from the judging dimensions and criteria shown in **Figure 13 and Figure 14**. We agree that this connection was not made explicit enough in the current draft, and we will add clearer discussion and citations to MQM and its derivatives in the revised version.
> > >
> > > We also note that our additional experiments on **Gemini** and **Qwen3-32B** with an **MQM-style prompt** already serve as a direct comparison to MQM-based evaluation. Specifically, **Gemini + MQM-style prompting achieves 89% accuracy**, while **Qwen3-32B + the same MQM-style prompting achieves 78% accuracy**. In comparison, our trained **JUDGER reaches 86% accuracy**. We believe these results show that:
> > >
> > > (1) MQM-style prompting is indeed a strong foundation, and
> > >
> > > (2) JUDGER’s improvement is not merely due to prompt wording, but comes from **task-specific training for cultural effectiveness evaluation**.
> > >
> > > We will make this relationship to MQM-based methods clearer in the revised paper and strengthen the corresponding discussion in the metric-comparison part.
> > >
> > > ### **2. On cross-lingual filtering and the concern of filtering out culturally local content**
> > >
> > > We would like to emphasize one important point: our benchmark pipeline does not stop at filtering. As shown in Figure 10, after the initial filtering stage, **we further use Gemini to enhance culturally salient characteristics, and then manually inspect and refine all 1,002 benchmark instances one by one**. Therefore, even if some cultural signals may be weakened during filtering, they are subsequently restored and strengthened in the final benchmark construction process.
> > >
> > > We will clarify this part more explicitly in the revised version, so that the benchmark construction process is not misunderstood as relying only on the filtering stage.
> > >
> > > ---
> > >
> > > Finally, thank you also for pointing out the typo in Line 187. We will correct the spelling error and revise the currently insufficient descriptions in the new version.
> > >
> > > We again sincerely thank the reviewer for the careful reading and constructive feedback. **Should you find that our revisions and clarifications have resolved your remaining concerns, we would greatly appreciate your reconsideration of the score.**

---

### Decision · Program_Chairs · 2026-04-30

**Decision:**

Accept (regular)

**Comment:**

This paper introduces a Chinese to English translation benchmark spanning with a focus on content styles (like social media slang) which can help ascertain the cultural effectiveness of the translation in addition to the translation accuracy. They also introduced a new metric JUDGER to measure cultural appropriateness of translation.

This is a great contribution for machine translation for user generated contents, reviewers are mostly concerned by the JUDGER and the weak baselines used in comparison with the new metric. However, the authors clarified that the JUDGER is based on Gemini distilled data and it has strong agreement with human annotations (Cohen’s κ = 0.72).